# LEARNING DIRECTED GRAPHICAL MODELS WITH OPTIMAL TRANSPORT

## ABSTRACT

Estimating the parameters of a probabilistic directed graphical model from incomplete data remains a long-standing challenge. This is because, in the presence of latent variables, both the likelihood function and posterior distribution are intractable without further assumptions about structural dependencies or model classes. While existing learning methods are fundamentally based on likelihood maximization, here we offer a new view of the parameter learning problem through the lens of optimal transport. This perspective licenses a general framework that operates on any directed graphs without making unrealistic assumptions on the posterior over the latent variables or resorting to black-box variational approximations. We develop a theoretical framework and support it with extensive empirical evidence demonstrating the flexibility and versatility of our approach. Across experiments, we show that not only can our method recover the ground-truth parameters but it also performs comparably or better on downstream applications, notably the non-trivial task of discrete representation learning.

## 1 INTRODUCTION

Learning probabilistic directed graphical models (DGMs, also known as Bayesian networks) with latent variables is an important ongoing challenge in machine learning and statistics. This paper focuses on parameter learning, i.e., estimating the parameters of a DGM given its known structure. Learning DGMs has a long history, dating back to classical indirect likelihood-maximization approaches such as expectation maximization (EM, Dempster et al., 1977). However, despite all its success stories, EM is well-known to suffer from local optima issues. More importantly, EM becomes inapplicable when the posterior distribution is intractable, which arises fairly often in practice.

A large family of related methods based on variational inference (VI, Jordan et al., 1999; Hoffman et al., 2013) have demonstrated tremendous potential in this case, where the evidence lower bound (ELBO) is not only used for posterior approximation but also for point estimation of the model parameters. Such an approach has proved surprisingly effective and robust to overfitting, especially when having a small number of parameters. From a high-level perspective, both EM and VI are based on likelihood maximization in the presence of latent variables, which ultimately requires carrying out expectations over the commonly intractable posterior. In order to address this challenge, a large spectrum of methods have been proposed in the literature and we refer the reader to Ambrogioni et al. (2021) for an excellent discussion of these approaches. Here we characterize them between two extremes. At one extreme, restrictive assumptions about the structure (e.g., as in mean-field approximations) or the model class (e.g., using conjugate exponential families) must be made to simplify the task. At the other extreme, when no assumptions are made, most existing black-box methods exploit very little information about the structure of the known probabilistic model (e.g., in black-box and stochastic VI (Ranganath et al., 2014; Hoffman et al., 2013), hierarchical approaches (Ranganath et al., 2016) or normalizing flows (Papamakarios et al., 2021)). Recently, VI has taken a significant leap forward by embracing amortized inference (Amos, 2022), which allows black-box optimization to be done in a considerably more efficient way.

Since the ultimate goal of VI is posterior inference, parameter estimation has been treated as a by-product of the optimization process where the model parameters are jointly updated with the variational parameters. As the complexity of the graph increases, despite the current advancements, parameter estimation in VI becomes less straightforward and computationally challenging.

Bridging this gap, we propose a scalable framework dedicated to learning parameters of a general directed graphical model. This alternative strategy inherits the flexibility of amortized optimization while eliminating the need to estimate expectations over the posterior distribution. Concretely, parameter learning is now viewed through the lens of *optimal transport* (Villani et al., 2009), where the data distribution is the source and the true model distribution is the target. Instead of minimizing a Kullback–Leibler (KL) divergence (which likelihood maximization methods are essentially doing), we aim to find a point estimate $\theta^*$ that minimizes the Wasserstein (WS) distance (Kantorovich, 1960) between these two distributions.

This perspective allows us to leverage desirable properties of WS distance in comparison with other metrics. These properties have motivated the recent surge in generative models, e.g., Wasserstein GANs (Adler & Lunz, 2018; Arjovsky et al., 2017) and Wasserstein Auto-encoders (WAE, Tolstikhin et al., 2017). Indeed, WS distance is shown to be well-behaved in situations where standard metrics such as the KL or JS (Jensen-Shannon) divergences are either infinite or undefined (Peyré et al., 2017; Ambrogioni et al., 2018). WS distance thus characterizes a more meaningful distance, especially when the two distributions reside in low-dimensional manifolds (Arjovsky et al., 2017).

Interestingly, akin to how Variational Auto-encoders (VAE, Kingma & Welling, 2013) is related to VI, our framework can be viewed as an extension of WAE for learning the parameters of a directed graphical model that can effectively exploit its structure. The parameter learning landscape is summarized in Figure 1.

**Contributions.** We present an entirely different view that casts parameter estimation as an optimal transport problem (Villani et al., 2009), where the goal is to find the optimal plan transporting "mass" from the data distribution to the model distribution. This permits a flexible framework applicable to any type of variable and graphical structure. In summary, we make the following contributions:

- We introduce **OTP-DAG** - an **O**ptimal **T**ransport framework for **P**arameter Learning in **D**irected **A**cyclic **G**raphical models[1]. OTP-DAG is an alternative line of thinking about parameter learning. Diverging from the existing frameworks, the underlying idea is to find the parameter set associated with the distribution that yields the lowest transportation cost from the data distribution.

- We present theoretical developments showing that minimizing the transport cost is equivalent to minimizing the reconstruction error between the observed data and the model generation. This renders a tractable training objective to be solved efficiently with stochastic gradient descent.

- We provide empirical evidence demonstrating the versatility of our method on various graphical structures. OTP-DAG is shown to successfully recover the ground-truth parameters and achieve comparable or better performance than competing methods across a range of downstream applications.

## 2 PRELIMINARIES

We first introduce the notations and basic concepts used throughout the paper. We reserve bold capital letters (i.e., $\mathbf{G}$) for notations related to graphs. We use calligraphic letters (i.e. $\mathcal{X}$) for spaces, italic capital letters (i.e. $X$) for random variables, and lower case letters (i.e. $x$) for their values.

A **directed graph** $\mathbf{G} = (\mathbf{V}, \mathbf{E})$ consists of a set of nodes $\mathbf{V}$ and an edge set $\mathbf{E} \subseteq \mathbf{V}^2$ of ordered pairs of nodes with $(v, v) \notin \mathbf{E}$ for any $v \in \mathbf{V}$ (one without self-loops). For a pair of nodes $i, j$ with $(i, j) \in \mathbf{E}$, there is an arrow pointing from $i$ to $j$ and we write $i \rightarrow j$. Two nodes $i$ and $j$ are adjacent if either $(i, j) \in \mathbf{E}$ or $(j, i) \in \mathbf{E}$. If there is an arrow from $i$ to $j$ then $i$ is a parent of $j$ and $j$ is a child of $i$. A Bayesian network structure $\mathbf{G} = (\mathbf{V}, \mathbf{E})$ is a **directed acyclic graph** (DAG), in which the nodes represent random variables

---

[1]Our code is anonymously published at https://anonymous.4open.science/r/OTP-7944/.

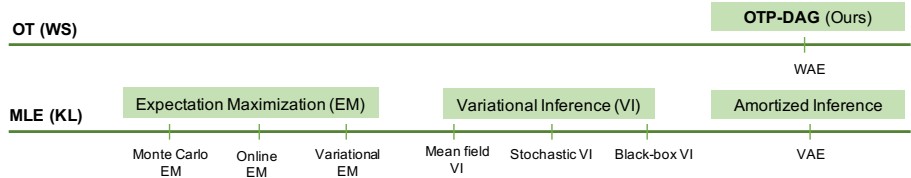

Figure 1: Notable parameter learning methods along the two lines of approaches. OTP-DAG can be viewed as an extension of WAE for learning parameters of a general directed graph, laying a foundation stone for a new paradigm of learning and, potentially, inference of graphical models.

$X = [X_i]_{i=1}^n$ with index set $\mathbf{V} := \{1, ..., n\}$. Let $\mathrm{PA}_{X_i}$ denote the set of variables associated with parents of node $i$ in $\mathbf{G}$. In this work, we tackle the classic yet important problem of learning the parameters of a directed graph from *partially observed data*. Let $\mathbf{O} \subseteq \mathbf{V}$ and $X_{\mathbf{O}} = [X_i]_{i \in \mathbf{O}}$ be the set of observed nodes and $\mathbf{H} := \mathbf{V} \backslash \mathbf{O}$ be the set of hidden nodes. Let $P_\theta$ and $P_d$ respectively denote the distribution induced by the graphical model and the empirical one induced by the *complete* (yet unknown) data. Given a fixed graphical structure $\mathbf{G}$ and some set of i.i.d data points, we aim to find the point estimate $\theta^*$ that best fits the observed data $X_{\mathbf{O}}$. The conventional approach is to minimize the KL divergence between the model distribution and the *empirical* data distribution over observed data i.e., $D_{\mathrm{KL}}(P_d(X_{\mathbf{O}}), P_\theta(X_{\mathbf{O}}))$, which is equivalent to maximizing the likelihood $P_\theta(X_{\mathbf{O}})$ w.r.t $\theta$. In the presence of latent variables, the marginal likelihood, given as $P_\theta(X_{\mathbf{O}}) = \int_{X_{\mathbf{H}}} P_\theta(X) dX_{\mathbf{H}}$, is generally intractable. Standard approaches then resort to maximizing a bound on the marginal log-likelihood, known as the evidence lower bound (ELBO), which is essentially the objective of EM (Moon, 1996) and VI (Jordan et al., 1999). Optimization of the ELBO for parameter learning in practice requires many considerations. We refer readers to Appendix B for a review of these intricacies.

## 3    OPTIMAL TRANSPORT FOR LEARNING DIRECTED GRAPHICAL MODELS

We begin by explaining how parameter learning can be reformulated into an optimal transport problem Villani (2003) and thereafter introduce our novel theoretical contribution.

We consider a DAG $\mathbf{G}(\mathbf{V}, \mathbf{E})$ over random variables $X = [X_i]_{i=1}^n$ that represents the data generative process of an underlying system. The system consists of $X$ as the set of endogenous variables and $U = \{U_i\}_{i=1}^n$ as the set of exogenous variables representing external factors affecting the system. Associated with every $X_i$ is an exogenous variable $U_i$ whose values are sampled from a prior distribution $P(U)$ independently from the other exogenous variables. For the purpose of theoretical development, our framework operates on an extended graph consisting of both endogenous and exogenous nodes (See Figure 2b). In the graph $\mathbf{G}$, $U_i$ is represented by a node with no ancestors that has an outgoing arrow towards node $i$. Every distribution $P_{\theta_i}(X_i | \mathrm{PA}_{X_i})$ henceforth can be reparameterized into a deterministic assignment

$$X_i = \psi_i(\mathrm{PA}_{X_i}, U_i), \text{ for } i = 1, ..., n.$$

The ultimate goal is to estimate $\theta = \{\theta_i\}_{i=1}^n$ as the parameters of the set of deterministic functions $\psi = \{\psi_i\}_{i=1}^n$. We will use the notation $\psi_\theta$ to emphasize this connection from now on. Given the empirical data distribution $P_d(X_{\mathbf{O}})$ and the model distribution $P_\theta(X_{\mathbf{O}})$ over the observed set $\mathbf{O}$, the **optimal transport** (OT) goal is to find the parameter set $\theta$ that minimizes the cost of transport between these two distributions. The Kantorovich's formulation of the problem is given by

$$W_c(P_d; P_\theta) := \inf_{\Gamma \sim \mathcal{P}(X \sim P_d, Y \sim P_\theta)} \mathbb{E}_{(X,Y) \sim \Gamma}[c(X, Y)], \tag{1}$$

where $\mathcal{P}(X \sim P_d, Y \sim P_\theta)$ is a set of all joint distributions of $(P_d; P_\theta)$; $c : \mathcal{X}_{\mathbf{O}} \times \mathcal{X}_{\mathbf{O}} \mapsto \mathcal{R}_+$ is any measurable cost function over $\mathcal{X}_{\mathbf{O}}$ (i.e., the product space of the spaces of observed variables) defined as $c(X_{\mathbf{O}}, Y_{\mathbf{O}}) := \sum_{i \in \mathbf{O}} c_i(X_i, Y_i)$ where $c_i$ is a measurable cost function over a space of an observed variable. Let $P_\theta(\mathrm{PA}_{X_i}, U_i)$ denote the joint distribution of $\mathrm{PA}_{X_i}$ and $U_i$ factorized according to the graphical model. Let $\mathcal{U}_i$ denote the space over random variable $U_i$. The key ingredient of our theoretical development is local backward mapping. For every observed node $i \in \mathbf{O}$, we define a stochastic "backward" map $\phi_i : \mathcal{X}_i \mapsto \Pi_{k \in \mathrm{PA}_{X_i}} \mathcal{X}_k \times \mathcal{U}_i$ such that $\phi_i \in \mathfrak{C}(X_i)$ where $\mathfrak{C}(X_i)$ is the constraint set given as

$$\mathfrak{C}(X_i) := \left\{ \phi_i : \phi_i \# P_d(X_i) = P_\theta(\mathrm{PA}_{X_i}, U_i) \right\};$$

that is, every backward $\phi_i \#$ defines a push forward operator such that the samples from $\phi_i(X_i)$ follow the marginal distribution $P_\theta(\mathrm{PA}_{X_i}, U_i)$.

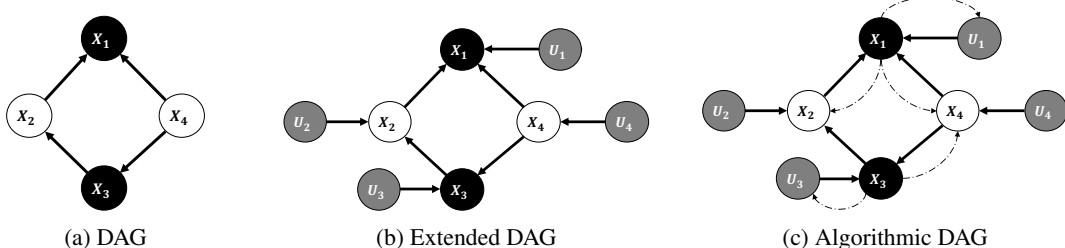

(a) DAG      (b) Extended DAG      (c) Algorithmic DAG

Figure 2: (a) A DAG represents a system of 4 endogenous variables where $X_1, X_3$ are observed (black-shaded) and $X_2, X_4$ are hidden variables (non-shaded). (b): The extended DAG includes an additional set of independent exogenous variables $U_1, U_2, U_3, U_4$ (grey-shaded) acting on each endogenous variable. $U_1, U_2, U_3, U_4 \sim P(U)$ where $P(U)$ is a prior product distribution. (c) Visualization of our backward-forward algorithm, where the dashed arcs represent the backward maps involved in optimization.

Theorem 1 presents the main theoretical contribution of our paper. Our OT problem seeks to find the optimal set of deterministic "forward" maps $\psi_\theta$ and stochastic "backward" maps $\left\{ \phi_i \in \mathfrak{C}(X_i) \right\}_{i \in \mathbf{O}}$ that minimize the cost of transporting the mass from $P_d$ to $P_\theta$ over $\mathbf{O}$. While the formulation in Eq. (1) is not trainable, we show that the problem is reduced to minimizing the reconstruction error between the data generated from $P_\theta$ and the observed data. To understand how reconstruction works, let us examine Figure 2c.

With a slight abuse of notations, for every $X_i$, we extend its parent set $\mathrm{PA}_{X_i}$ to include an exogenous variable and possibly some other endogenous variables. Given $X_1$ and $X_3$ as observed nodes, we first sample $X_1 \sim P_d(X_1), X_3 \sim P_d(X_3)$ and evaluate the local densities $P_{\phi_1}(\mathrm{PA}_{X_1}|X_1)$, $P_{\phi_3}(\mathrm{PA}_{X_3}|X_3)$ where $\mathrm{PA}_{X_1} = \{X_2, X_4, U_1\}$ and $\mathrm{PA}_{X_3} = \{X_4, U_3\}$. The next step is to sample $\mathrm{PA}_{X_1} \sim P_{\phi_1}(\mathrm{PA}_{X_1}|X_1)$ and $\mathrm{PA}_{X_3} \sim P_{\phi_3}(\mathrm{PA}_{X_3}|X_3)$, which are plugged back to the model $\psi_\theta$ to obtain the reconstructions $\widetilde{X_1} = \psi_{\theta_1}(\mathrm{PA}_{X_1})$ and $\widetilde{X_3} = \psi_{\theta_3}(\mathrm{PA}_{X_3})$. We wish to learn $\theta$ such that $X_1$ and $X_3$ are reconstructed correctly. For a general graphical model, this optimization objective is formalized as

**Theorem 1.** *For every $\phi_i$ as defined above and fixed $\psi_\theta$,*

$$W_c\big(P_d(X_{\mathbf{O}}); P_\theta(X_{\mathbf{O}})\big) = \inf_{\left[\phi_i \in \mathfrak{C}(X_i)\right]_{i \in \mathbf{O}}} \mathbb{E}_{X_{\mathbf{O}} \sim P_d(X_{\mathbf{O}}), \mathrm{PA}_{X_{\mathbf{O}}} \sim \phi(X_{\mathbf{O}})} \big[ c\big(X_{\mathbf{O}}, \psi_\theta(\mathrm{PA}_{X_{\mathbf{O}}})\big) \big], \quad (2)$$

*where* $\mathrm{PA}_{X_{\mathbf{O}}} := \big[ [X_{ij}]_{j \in \mathrm{PA}_{X_i}} \big]_{i \in \mathbf{O}}$.

The proof is provided in Appendix A. While Theorem 1 set ups a tractable form for our optimization solution, the quality of the reconstruction hinges on how well the backward maps approximate the true local densities. To ensure approximation fidelity, every backward function $\phi_i$ must satisfy its push-forward constraint defined by $\mathfrak{C}$. In the above example, the backward maps $\phi_i$ and $\phi_3$ must be constructed such that

$\phi_1 \# P_d(X_1) = P_\theta(X_2, X_4, U_1)$ and $\phi_3 \# P_d(X_3) = P_\theta(X_4, U_3)$. This results in a constraint optimization problem, and we relax the constraints by adding a penalty to the above objective.

The **final optimization objective** is therefore given as

$$J_{\text{WS}} = \inf_{\psi, \phi} \; \mathbb{E}_{X_{\mathbf{O}} \sim P_d(X_{\mathbf{O}}), \text{PA}_{X_{\mathbf{O}}} \sim \phi(X_{\mathbf{O}})} \big[ c\big(X_{\mathbf{O}}, \psi_\theta(\text{PA}_{X_{\mathbf{O}}})\big) \big] + \eta \, D\big(P_\phi, P_\theta\big), \tag{3}$$

where $D$ is any arbitrary divergence measure and $\eta > 0$ is a trade-off hyper-parameter. $D\big(P_\phi, P_\theta\big)$ is a short-hand for divergence between all pairs of backward and forward distributions.

**Connection with Auto-encoders.** OTP-DAG is an optimization-based approach in which we leverage reparameterization and amortized inference (Amos, 2022) for solving it efficiently via stochastic gradient descent. This theoretical result provides us with two interesting properties: **(1)** all model parameters are optimized simultaneously within a single framework whether the variables are continuous or discrete, and **(2)** the computational process can be automated without the need for analytic lower bounds (as in EM and traditional VI), specific graphical structures (as in mean-field VI), or priors over variational distributions on latent variables (as in hierarchical VI). The flexibility our method exhibits is akin to VAE, and OTP-DAG in fact serves as an extension of WAE for learning general directed graphical models. Our formulation thus inherits a desirable characteristic from that of WAE, which specifically helps mitigate the posterior collapse issue notoriously occurring to VAE. Appendix D explains this in more detail. Particularly, in the next section, we will empirically show that OTP-DAG effectively alleviates the codebook collapse issue in discrete representation learning. Details on our algorithm can be found in Appendix C.

## 4 APPLICATIONS

In this section, we illustrate the practical application of the OTP-DAG algorithm. Instead of achieving state-of-the-art performance on specific applications, our key objective is to demonstrate the versatility of OTP-DAG: our method can be harnessed for a wide range of purposes in a single learning procedure. In terms of estimation accuracy, OTP-DAG is capable of recovering the ground-truth parameters while achieving the comparable or better performance level of existing frameworks across downstream tasks.

**Experimental setup.** We consider various directed probabilistic models with either continuous or discrete variables. We begin with (1) Latent Dirichlet Allocation Blei et al. (2003) for topic modeling and (2) Hidden Markov Model (HMM) for sequential modeling. We conclude with a more challenging setting: (3) Discrete Representation Learning (Discrete RepL) that cannot simply be solved by EM or MAP (maximum a posteriori). It in fact invokes deep generative modeling via a pioneering development called Vector Quantization Variational Auto-Encoder (VQ-VAE, Van Den Oord et al., 2017). We attempt to apply OTP-DAG for learning discrete representations by grounding it into a parameter learning problem. Figure 3 illustrates the empirical DAG structures of 3 applications. Unlike the standard visualization where the parameters are considered hidden nodes, our graph separates model parameters from latent variables and only illustrates random variables and their dependencies (except the special setting of Discrete RepL). We also omit the exogenous variables associated with the hidden nodes for visibility, since only those acting on the observed nodes are relevant for computation. There is also a noticeable difference between Figure 3 and Figure 2c: the empirical version does not require learning the backward maps for the exogenous variables. It is observed across our experiments that sampling the noise from an appropriate prior distribution suffices to yield accurate estimation, which is in fact beneficial in that training time can be greatly reduced.

**Baselines.** We compare OTP-DAG with two groups of parameter learning methods towards the two extremes: (1) MAP, EM and SVI where analytic derivation is required; (2) variational auto-encoding frameworks (the closest baseline to ours) where black-box optimization is permissible. For the latter, we here report the performance of vanilla VAE-based models, while providing additional experiments with some advances in Appendix E. We also leave the discussion of the formulation and technicalities in Appendix E. In all tables, we report the average results over 5 random initializations and the best/second-best ones are bold/underlined. In addition, $\uparrow, \downarrow$ indicate higher/lower performance is better, respectively.

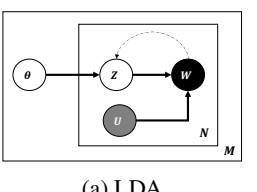

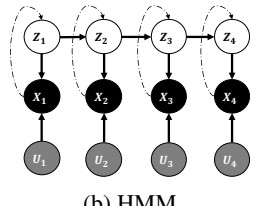

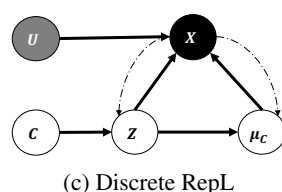

(a) LDA  (b) HMM  (c) Discrete RepL

Figure 3: Empirical structures of (a) latent Dirichlet allocation model (in plate notation), (b) standard hidden Markov model, and (c) discrete representation learning.

### 4.1 LATENT DIRICHLET ALLOCATION

Let us consider a corpus $\mathcal{D}$ of $M$ independent documents where each document is a sequence of $N$ words denoted by $W = (W_1, W_2, \cdots, W_N)$. Documents are represented as random mixtures over $K$ latent topics, each of which is characterized by a distribution over words. Let $V$ be the size of a vocabulary indexed by $\{1, \cdots, V\}$. Latent Dirichlet Allocation (LDA) (Blei et al., 2003) dictates the following generative process for every document in the corpus:

1. Sample $\theta \sim \text{Dir}(\alpha)$ with $\alpha < 1$,
2. Sample $\gamma_k \sim \text{Dir}(\beta)$ where $k \in \{1, \cdots, K\}$,
3. For each of the word positions $n \in \{1, \cdots, N\}$,
   - Sample a topic $Z_n \sim \text{Multi-nominal}(\theta)$,
   - Sample a word $W_n \sim \text{Multi-nominal}(\gamma_k)$,

where $\text{Dir}(.)$ is a Dirichlet distribution. $\theta$ is a $K-$dimensional vector that lies in the $(K-1)-$simplex and $\gamma_k$ is a $V-$dimensional vector represents the word distribution corresponding to topic $k$. In the standard model, $\alpha, \beta, K$ are hyper-parameters and $\theta, \gamma$ are learnable parameters. Throughout the experiments, the number of topics $K$ is assumed known and fixed.

**Parameter Estimation.** To test whether OTP-DAG can recover the true parameters, we generate synthetic data in the setting: the word probabilities are parameterized by a $K \times V$ matrix $\gamma$ where $\gamma_{kn} := P(W_n = 1 | Z_n = 1)$; $\gamma$ is now a fixed quantity to be estimated. We set $\alpha = 1/K$ uniformly and generate small datasets for different number of topics $K$ and sample size $N$. Following Griffiths & Steyvers (2004), for every topic $k$, the word distribution $\gamma_k$ can be represented as a square grid where each cell, corresponding to a word, is assigned an integer value of either $0$ and $1$, indicating whether a certain word is allocated to the $k^{th}$ topic or not. As a result, each topic is associated with a specific pattern. For simplicity, we represent topics using horizontal or vertical patterns (See Figure 4a). According to the above generative model, we sample data w.r.t 3 sets of configuration triplets $\{K, M, N\}$. We compare OTP-DAG with Batch EM and SVI and Prod LDA - a variational auto-encoding topic model (Srivastava & Sutton, 2017).

Table 1: Fidelity of estimates of the topic-word distribution $\gamma$ across 3 settings. Fidelity is measured by KL divergence, Hellinger (HL) (Hellinger, 1909) and Wasserstein distance with the ground-truth distributions.

| Metric ↓ | $K$ | $M$ | $N$ | **OTP-DAG** (Ours) | **Batch EM** | **SVI** | **Prod LDA** |
|---|---|---|---|---|---|---|---|
| HL | 10 | 1,000 | 100 | **2.327 ± 0.009** | 2.807 ± 0.189 | 2.712 ± 0.087 | 2.353 ± 0.012 |
| KL | 10 | 1,000 | 100 | 1.701 ± 0.005 | 1.634 ± 0.022 | **1.602 ± 0.014** | 1.627 ± 0.027 |
| WS | 10 | 1,000 | 100 | **0.027 ± 0.004** | 0.058 ± 0.000 | 0.059 ± 0.000 | 0.052 ± 0.001 |
| HL | 20 | 5,000 | 200 | 3.800 ± 0.058 | 4.256 ± 0.084 | 4.259 ± 0.096 | **3.700 ± 0.012** |
| KL | 20 | 5,000 | 200 | 2.652 ± 0.080 | **2.304 ± 0.004** | 2.305 ± 0.003 | 2.316 ± 0.026 |
| WS | 20 | 5,000 | 200 | **0.010 ± 0.001** | 0.022 ± 0.000 | 0.022 ± 0.001 | 0.018 ± 0.000 |
| HL | 30 | 10,000 | 300 | 4.740 ± 0.029 | 5.262 ± 0.077 | 5.245 ± 0.035 | **4.723 ± 0.017** |
| KL | 30 | 10,000 | 300 | 2.959 ± 0.015 | **2.708 ± 0.002** | 2.709 ± 0.001 | 2.746 ± 0.034 |
| WS | 30 | 10,000 | 300 | **0.005 ± 0.001** | 0.012 ± 0.000 | 0.012 ± 0.000 | 0.009 ± 0.000 |

Table 1 reports the fidelity of the estimation of $\gamma$. OTP-DAG consistently achieves high-quality estimates by both Hellinger and Wasserstein distances. It is straightforward that the baselines are superior by the KL metric, as it is what they implicitly minimize. While it is inconclusive from the numerical estimations, the qualitative results complete the story. Figure 4a illustrates the distributions of individual words to the topics from each method after 300 training epochs. OTP-DAG successfully recovers the true patterns and as well as EM and SVI. More qualitative examples for the other settings are presented in Figures 7 and 8 where OTP-DAG is shown to recover almost all true patterns.

**Topic Inference.** We now demonstrate the effectiveness of OTP-DAG on downstream applications. We here use OTP-DAG to infer the topics of 3 real-world datasets:[2] 20 News Group, BBC News and DBLP. We revert to the original generative process where the topic-word distribution follows a Dirichlet distribution parameterized by the concentration parameters $\beta$, instead of having $\gamma$ as a fixed quantity. $\beta$ is now initialized as a matrix of real values $\left(\beta \in \mathbb{R}^{K \times V}\right)$ representing the log concentration values.

Table 5 reports the quality of the inferred topics, which is evaluated via the diversity and coherence of the selected words. Diversity refers to the proportion of unique words, whereas Coherence is measured with normalized pointwise mutual information (Aletras & Stevenson, 2013), reflecting the extent to which the words in a topic are associated with a common theme. There exists a trade-off between Diversity and Coherence: words that are excessively diverse greatly reduce coherence, while a set of many duplicated words yields higher coherence yet harms diversity. A well-performing topic model would strike a good balance between these metrics. If we consider two metrics comprehensively, our method achieves comparable or better performance than the other learning algorithms

### 4.2 HIDDEN MARKOV MODELS

This application deals with time-series data following a **Poisson hidden Markov model** (See Figure 3b). Given a time series of $T$ steps, the task is to segment the data stream into $K$ different states, each of which follows a Poisson distribution with rate $\lambda_k$. The observation at each step $t$ is given as

$$P(X_t|Z_t = k) = \text{Poi}(X_t|\lambda_k), \quad \text{for } k = 1, \cdots, K.$$

Following Murphy (2023), we use a uniform prior over the initial state. The Markov chain stays in the current state with probability $p$ and otherwise transitions to one of the other $K-1$ states uniformly at random. The transition distribution is given as

$$Z_1 \sim \text{Cat}\left(\left\{\frac{1}{4}, \frac{1}{4}, \frac{1}{4}, \frac{1}{4}\right\}\right), \quad Z_t|Z_{t-1} \sim \text{Cat}\left(\left\{\begin{array}{ll} p & \text{if } Z_t = Z_{t-1} \\ \frac{1-p}{4-1} & \text{otherwise} \end{array}\right\}\right)$$

Let $P(Z_1)$ and $P(Z_t|Z_{t-1})$ respectively denote these prior transition distributions. We generate a synthetic dataset $\mathcal{D}$ of 200 observations at rates $\lambda = \{12, 87, 60, 33\}$ with change points occurring at times $(40, 60, 55)$. We would like to learn the concentration parameters $\lambda_{1:K} = [\lambda_k]_{k=1}^K$ through which segmentation can be realized, assuming that the number of states $K = 4$ is known.

The true transition probabilities are generally unknown. The value $p$ is treated as a hyper-parameter and we fit HMM with 6 choices of $p$. Table 2 demonstrates the quality of our estimates, in comparison with MAP estimates. Our estimation approaches the ground-truth values comparably to MAP. We note that the MAP solution requires the analytical marginal likelihood of the model, which is not necessary for our method. Figure 4b reports the most probable state for each observation, inferred from our backward distribution $\phi(X_{1:T})$. It can be seen that the partition overall aligns with the true generative process of the data.

By observing the data, one can assume $p$ should be relatively high, $0.75 - 0.95$ seems most reasonable. This explains why the MAP estimation at $p = 0.05$ is terrible. Meanwhile, for our OTP-DAG, the effect of $p$ is controlled by the trade-off coefficient $\eta$. We here fix $\eta = 0.1$. Since the effect is fairly minor, OTP-DAG estimates across $p$ are less variant. Table 7 additionally analyzes the model performance when $\eta$ varies.

---

[2] https://github.com/MIND-Lab/OCTIS.

Table 2: Estimates of $\lambda_{1:4}$ at various transition probabilities $p$ and mean absolute reconstruction error.

| $p$ | $\lambda_1 = 12$ | $\lambda_2 = 87$ | $\lambda_3 = 60$ | $\lambda_4 = 33$ | $\lambda_1 = 12$ | $\lambda_2 = 87$ | $\lambda_3 = 60$ | $\lambda_4 = 33$ |
|---|---|---|---|---|---|---|---|---|
| | **OTP-DAG Estimates** (Ours) | | | | **MAP Estimates** | | | |
| 0.05 | 11.83 | 87.20 | 60.61 | 33.40 | 14.88 | 85.22 | 71.42 | 40.39 |
| 0.15 | 11.62 | 87.04 | 59.69 | 32.85 | 12.31 | 87.11 | 61.86 | 33.90 |
| 0.35 | 11.77 | 86.76 | 60.01 | 33.26 | 12.08 | 87.28 | 60.44 | 33.17 |
| 0.55 | 11.76 | 86.98 | 60.15 | 33.38 | 12.05 | 87.12 | 60.12 | 33.01 |
| 0.75 | 11.63 | 86.46 | 60.04 | 33.57 | 12.05 | 86.96 | 59.98 | 32.94 |
| 0.95 | 11.57 | 86.92 | 60.36 | 33.06 | 12.05 | 86.92 | 59.94 | 32.93 |
| MAE $\downarrow$ | **0.30** | **0.19** | **0.25** | **0.30** | 0.57 | 0.40 | 2.32 | 1.43 |

(a) LDA topic modeling

(b) Poisson time-series segmentation

Figure 4: (a) The topic-word distributions recovered from each method after 300-epoch training. (b) Segmentation of Poisson time series inferred from the backward distribution $\phi(X_{1:T})$.

## 4.3 LEARNING DISCRETE REPRESENTATIONS

Many types of data exist in discrete symbols e.g., words in texts, or pixels in images. This motivates the need to explore the latent discrete representations of the data, which can be useful for planning and symbolic reasoning tasks. Viewing discrete representation learning as a parameter learning problem, we endow it with a probabilistic generative process as illustrated in Figure 3c. The problem deals with a latent space $\mathcal{C} \in \mathbb{R}^{K \times D}$ composed of $K$ discrete latent sub-spaces of $D$ dimensionality. The probability a data point belongs to a discrete sub-space $c \in \{1, \cdots, K\}$ follows a $K-$way categorical distribution $\pi = [\pi_1, \cdots, \pi_K]$. In the language of VQ-VAE, each $c$ is referred to as a *codeword* and the set of codewords is called a *codebook*. Let $Z \in \mathbb{R}^D$ denote the latent variable in a sub-space. On each sub-space, we impose a Gaussian distribution parameterized by $\mu_c, \Sigma_c$ where $\Sigma_c$ is diagonal. The data generative process is described as follows:

1. Sample $c \sim \text{Cat}(\pi)$ and $Z \sim \mathcal{N}(\mu_c, \Sigma_c)$
2. Quantize $\mu_c = Q(Z)$,
3. Generate $X = \psi_\theta(Z, \mu_c)$.

where $\psi$ is a highly non-convex function with unknown parameters $\theta$ and often parameterized with a deep neural network. $Q$ refers to the quantization of $Z$ to $\mu_c$ defined as $\mu_c = Q(Z)$ where $c = \text{argmin}_c d_z(Z; \mu_c)$ and $d_z = \sqrt{(Z - \mu_c)^T \Sigma_c^{-1}(Z - \mu_c)}$ is the Mahalanobis distance.

The goal is to learn the parameter set $\{\pi, \mu, \Sigma, \theta\}$ with $\mu = [\mu_k]_{k=1}^K, \Sigma = [\Sigma_k]_{k=1}^K$ such that the learned representation captures the key properties of the data. Following VQ-VAE, our practical implementation considers $Z$ as an $M-$component latent embedding. We experiment with images in this application and compare OTP-DAG with VQ-VAE on CIFAR10, MNIST, SVHN and CELEBA datasets. Since the true parameters are unknown, we assess how well the latent space characterizes the input data through the quality

of the reconstruction of the original images. Table 3 reports our superior performance in preserving high-quality information of the input images. VQ-VAE suffers from poorer performance mainly due to *codebook collapse* (Yu et al., 2021) where most of latent vectors are quantized to limited discrete codewords. Meanwhile, our framework allows for control over the number of latent representations, ensuring all codewords are utilized. In Appendix E.3, we detail the formulation of our method and provide qualitative examples. We also showcase therein our competitive performance against a recent advance called SQ-VAE (Takida et al., 2022) without introducing any additional complexity.

Table 3: Quality of the image reconstructions ($K = 512$).

| Dataset | Method | Latent Size | SSIM ↑ | PSNR ↑ | LPIPS ↓ | rFID ↓ | Perplexity ↑ |
|---|---|---|---|---|---|---|---|
| CIFAR10 | **VQ-VAE** | $8 \times 8$ | 0.70 | 23.14 | 0.35 | 77.3 | 69.8 |
| | **OTP-DAG** (Ours) | $8 \times 8$ | **0.80** | **25.40** | **0.23** | **56.5** | **498.6** |
| MNIST | **VQ-VAE** | $8 \times 8$ | **0.98** | 33.37 | 0.02 | 4.8 | 47.2 |
| | **OTP-DAG** (Ours) | $8 \times 8$ | **0.98** | **33.62** | **0.01** | **3.3** | **474.6** |
| SVHN | **VQ-VAE** | $8 \times 8$ | 0.88 | 26.94 | 0.17 | 38.5 | 114.6 |
| | **OTP-DAG** (Ours) | $8 \times 8$ | **0.94** | **32.56** | **0.08** | **25.2** | **462.8** |
| CELEBA | **VQ-VAE** | $16 \times 16$ | 0.82 | 27.48 | 0.19 | 19.4 | 48.9 |
| | **OTP-DAG** (Ours) | $16 \times 16$ | **0.88** | **29.77** | **0.11** | **13.1** | **487.5** |

## 5 DISCUSSION AND CONCLUSION

The key message across our experiments is that OTP-DAG is a scalable and versatile framework readily applicable to learning any directed graphs with latent variables. OTP-DAG is consistently shown to perform comparably and in some cases better than MAP, EM and SVI which are well-known for yielding reliable estimates. Similar to amortized VI, on one hand, our method employs amortized optimization and assumes one can sample from the priors or more generally, the model marginals over latent parents. OTP-DAG requires continuous relaxation through reparameterization of the underlying model distribution to ensure the gradients can be back-propagated effectively. The specification is also not unique to OTP-DAG: VAE also relies on reparameterization trick to compute the gradients w.r.t the variational parameters. For discrete distributions and for some continuous ones (e.g., Gamma distribution), this is not easy to attain. To this end, a proposal on *Generalized Reparameterization Gradient* (Ruiz et al., 2016) is a viable solution. On the other hand, different from VI, our global OT cost minimization is achieved by characterizing local densities through backward maps from the observed nodes to their parents. This localization strategy makes it easier to find a good approximation compared to VI, where the variational distribution is defined over all hidden variables and should ideally characterize the entire global dependencies in the graph. A popular method called Semi-amortized VAE (SA-VAE, Kim et al., 2018) is proposed to tackle this sub-optimality issue of the inference network in VI. In Appendix D, we compare OTP-DAG with this model on parameter estimation task, where ours competes on par with SA-VAE under the usual OTP-DAG learning procedure that comes with no extra overhead. To model the backward distributions, we utilize the expressivity of deep neural networks. Based on the universal approximation theorem (Hornik et al., 1989), the gap between the model distribution and the true conditional can be assumed to be smaller than an arbitrary constant $\epsilon$ given enough data, network complexity, and training time.

**Future Research.** The proposed algorithm lays the cornerstone for an exciting paradigm shift in the realm of graphical learning and inference. Looking ahead, this fresh perspective unlocks a wealth of promising avenues for future application of OTP-DAG to large-scale inference problems or other learning tasks such as for undirected graphical models, or structural learning where edge existence and directionality can be parameterized as part of the model parameters.

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

## A  ALL PROOFS

We now present the proof of Theorem 1 which is the key theorem in our paper.

**Theorem 1.** . *For every $\phi_i$ as defined above and fixed $\psi_\theta$,*

$$W_c\big(P_d(X_{\mathbf{O}}); P_\theta(X_{\mathbf{O}})\big) = \inf_{\left[\phi_i \in \mathfrak{C}(X_i)\right]_{i \in \mathbf{O}}} \mathbb{E}_{X_{\mathbf{O}} \sim P_d(X_{\mathbf{O}}), \mathrm{PA}_{X_{\mathbf{O}}} \sim \phi(X_{\mathbf{O}})}\big[c\big(X_{\mathbf{O}}, \psi_\theta(\mathrm{PA}_{X_{\mathbf{O}}})\big)\big],$$

*where* $\mathrm{PA}_{X_{\mathbf{O}}} := \big[[X_{ij}]_{j \in \mathrm{PA}_{X_i}}\big]_{i \in \mathbf{O}}.$

*Proof.* Let $\Gamma \in \mathcal{P}(P_d(X_{\mathbf{O}}), P_\theta(X_{\mathbf{O}}))$ be the optimal joint distribution over $P_d(X_{\mathbf{O}})$ and $P_\theta(X_{\mathbf{O}})$ of the corresponding Wasserstein distance. We consider three distributions: $P_d(X_{\mathbf{O}})$ over $A = \prod_{i \in \mathbf{O}} \mathcal{X}_i$, $P_\theta(X_{\mathbf{O}}))$ over $C = \prod_{i \in \mathbf{O}} \mathcal{X}_i$, and $P_\theta(\mathrm{PA}_{X_{\mathbf{O}}}) = P_\theta([\mathrm{PA}_{X_i}]_{i \in \mathbf{O}})$ over $B = \prod_{i \in \mathbf{O}} \prod_{k \in \mathrm{PA}_{X_i}} \mathcal{X}_k$. Here we note that the last distribution $P_\theta(\mathrm{PA}_{X_{\mathbf{O}}}) = P_\theta([\mathrm{PA}_{X_i}]_{i \in \mathbf{O}})$ is the model distribution over the parent nodes of the observed nodes.

It is evident that $\Gamma \in \mathcal{P}(P_d(X_{\mathbf{O}}), P_\theta(X_{\mathbf{O}}))$ is a joint distribution over $P_d(X_{\mathbf{O}})$ and $P_\theta(X_{\mathbf{O}})$; let $\beta = (id, \psi_\theta) \# P_\theta([\mathrm{PA}_{X_i}]_{i \in \mathbf{O}})$ be a deterministic coupling or joint distribution over $P_\theta([\mathrm{PA}_{X_i}]_{i \in \mathbf{O}})$ and $P_\theta(X_{\mathbf{O}})$. Using the gluing lemma (see Lemma 5.5 in Santambrogio (2015)), there exists a joint distribution $\alpha$ over $A \times B \times C$ such that $\alpha_{AC} = (\pi_A, \pi_C) \# \alpha = \Gamma$ and $\alpha_{BC} = (\pi_B, \pi_C) \# \alpha = \beta$ where $\pi$ is the projection operation. Let us denote $\gamma = (\pi_A, \pi_B) \# \alpha$ as a joint distribution over $P_d(X_{\mathbf{O}})$ and $P_\theta([\mathrm{PA}_{X_i}]_{i \in \mathbf{O}})$.

Given $i \in \mathbf{O}$, we denote $\gamma_i$ as the projection of $\gamma$ over $\mathcal{X}_i$ and $\prod_{k \in \mathrm{PA}_{X_i}} \mathcal{X}_k$. We further denote $\phi_i(X_i) = \gamma_i(\cdot \mid X_i)$ as a stochastic map from $\mathcal{X}_i$ to $\prod_{k \in \mathrm{PA}_{X_i}} \mathcal{X}_k$. It is worth noting that because $\gamma_i$ is a joint distribution over $P_d(X_i)$ and $P_\theta(\mathrm{PA}_{X_i})$, $\phi_i \in \mathfrak{C}(X_i)$.

$$\begin{aligned}
W_c\left(P_d\left(X_{\mathbf{O}}\right), P_\theta\left(X_{\mathbf{O}}\right)\right) &= \mathbb{E}_{\left(X_{\mathbf{O}}, \tilde{X}_{\mathbf{O}}\right) \sim \Gamma}\left[c\left(X_{\mathbf{O}}, \tilde{X}_{\mathbf{O}}\right)\right] = \mathbb{E}_{\left(X_{\mathbf{O}}, \mathrm{PA}_{X_{\mathbf{O}}}, \tilde{X}_{\mathbf{O}}\right) \sim \alpha}\left[c\left(X_{\mathbf{O}}, \tilde{X}_{\mathbf{O}}\right)\right] \\
&= \mathbb{E}_{X_{\mathbf{O}} \sim P_d, \left[\mathrm{PA}_{X_i} \sim \gamma_i(\cdot | X_i)\right]_{i \in \mathbf{O}}, \tilde{X}_{\mathbf{O}} \sim \alpha_{BC}(\cdot | \mathrm{PA}_{X_{\mathbf{O}}})}\left[c\left(X_{\mathbf{O}}, \tilde{X}_{\mathbf{O}}\right)\right] \\
&\overset{(1)}{=} \mathbb{E}_{X_{\mathbf{O}} \sim P_d, \left[\mathrm{PA}_{X_i} = \phi_i(X_i)\right]_{i \in \mathbf{O}}, \tilde{X}_{\mathbf{O}} = \psi_\theta(\mathrm{PA}_{X_{\mathbf{O}}})}\left[c\left(X_{\mathbf{O}}, \tilde{X}_{\mathbf{O}}\right)\right] \\
&= \mathbb{E}_{X_{\mathbf{O}} \sim P_d, \mathrm{PA}_{X_{\mathbf{O}}} = \phi(X_{\mathbf{O}}), \tilde{X}_{\mathbf{O}} = \psi_\theta(\mathrm{PA}_{X_{\mathbf{O}}})}\left[c\left(X_{\mathbf{O}}, \tilde{X}_{\mathbf{O}}\right)\right] \\
&\overset{(2)}{=} \mathbb{E}_{X_{\mathbf{O}} \sim P_d, \mathrm{PA}_{X_{\mathbf{O}}} = \phi(X_{\mathbf{O}})}\left[c\left(X_{\mathbf{O}}, \psi_\theta\left(\mathrm{PA}_{X_{\mathbf{O}}}\right)\right)\right] \\
&\geq \inf_{\left[\phi_i \in \mathfrak{C}(X_i)\right]_{i \in \mathbf{O}}} \mathbb{E}_{X_{\mathbf{O}} \sim P_d, \mathrm{PA}_{X_{\mathbf{O}}} = \phi(X_{\mathbf{O}})}\left[c\left(X_{\mathbf{O}}, \psi_\theta\left(\mathrm{PA}_{X_{\mathbf{O}}}\right)\right)\right]. \quad (4)
\end{aligned}$$

Here we note that we have $\overset{(1)}{=}$ because $\alpha_{BC}$ is a deterministic coupling and we have $\overset{(2)}{=}$ because the expectation is preserved through a deterministic push-forward map.

Let $[\phi_i \in \mathfrak{C}(X_i)]_{i \in \mathbf{O}}$ be the optimal backward maps of the optimization problem (OP) in (6). We define the joint distribution $\gamma$ over $P_d(X_{\mathbf{O}})$ and $P_\theta(\mathrm{PA}_{X_{\mathbf{O}}}) = P_\theta([\mathrm{PA}_{X_i}]_{i \in \mathbf{O}})$ as follows. We first sample $X_{\mathbf{O}} \sim P_d(X_{\mathbf{O}})$ and for each $i \in \mathbf{O}$, we sample $\mathrm{PA}_{X_i} \sim \phi_i(X_i)$, and finally gather $(X_{\mathbf{O}}, \mathrm{PA}_{X_{\mathbf{O}}}) \sim \gamma$ where $\mathrm{PA}_{X_{\mathbf{O}}} = [\mathrm{PA}_{X_i}]_{i \in \mathbf{O}}$. Consider the joint distribution $\gamma$ over $P_d(X_{\mathbf{O}}), P_\theta(\mathrm{PA}_{X_{\mathbf{O}}}) = P_\theta([\mathrm{PA}_{X_i}]_{i \in \mathbf{O}})$ and the deterministic coupling or joint distribution $\beta = (id, \psi_\theta) \# P_\theta([\mathrm{PA}_{X_i}]_{i \in \mathbf{O}})$ over $P_\theta([\mathrm{PA}_{X_i}]_{i \in \mathbf{O}})$ and $P_\theta(X_{\mathbf{O}})$, the gluing lemma indicates the existence of the joint distribution $\alpha$ over $A \times C \times B$ such that $\alpha_{AB} = (\pi_A, \pi_B) \# \alpha = \gamma$ and $\alpha_{BC} = (\pi_B, \pi_C) \# \alpha = \beta$. We further denote $\Gamma = \alpha_{AC} = (\pi_A, \pi_C) \# \alpha$ which is a

joint distribution over $P_d(X_{\mathbf{O}})$ and $P_\theta(X_{\mathbf{O}})$. It follows that

$$
\inf_{[\phi_i \in \mathfrak{C}(X_i)]_{i \in \mathbf{O}}} \mathbb{E}_{X_{\mathbf{O}} \sim P_d, \mathrm{PA}_{X_{\mathbf{O}}} = \phi(X_{\mathbf{O}})} \left[ c\left(X_{\mathbf{O}}, \psi_\theta\left(\mathrm{PA}_{X_{\mathbf{O}}}\right)\right) \right]
$$

$$
= \mathbb{E}_{X_{\mathbf{O}} \sim P_d, \mathrm{PA}_{X_{\mathbf{O}}} = \phi(X_{\mathbf{O}})} \left[ c\left(X_{\mathbf{O}}, \psi_\theta\left(\mathrm{PA}_{X_{\mathbf{O}}}\right)\right) \right]
$$

$$
\overset{(1)}{=} \mathbb{E}_{X_{\mathbf{O}} \sim P_d, \mathrm{PA}_{X_{\mathbf{O}}} \sim \phi(X_{\mathbf{O}}), \tilde{X}_{\mathbf{O}} = \psi_\theta\left(\mathrm{PA}_{X_{\mathbf{O}}}\right)} \left[ c\left(X_{\mathbf{O}}, \tilde{X}_{\mathbf{O}}\right) \right]
$$

$$
= \mathbb{E}_{X_{\mathbf{O}} \sim P_d, \mathrm{PA}_{X_{\mathbf{O}}} \sim \gamma(\cdot | X_{\mathbf{O}}), \tilde{X}_{\mathbf{O}} \sim \alpha_{BC}(\cdot | \mathrm{PA}_{X_{\mathbf{O}}})} \left[ c\left(X_{\mathbf{O}}, \tilde{X}_{\mathbf{O}}\right) \right]
$$

$$
= \mathbb{E}_{\left(X_{\mathbf{O}}, \mathrm{PA}_{X_{\mathbf{O}}}, \tilde{X}_{\mathbf{O}}\right) \sim \alpha} \left[ c\left(X_{\mathbf{O}}, \tilde{X}_{\mathbf{O}}\right) \right]
$$

$$
= \mathbb{E}_{\left(X_{\mathbf{O}}, \tilde{X}_{\mathbf{O}}\right) \sim \Gamma} \left[ c\left(X_{\mathbf{O}}, \tilde{X}_{\mathbf{O}}\right) \right] \geq W_c\left(P_d\left(X_{\mathbf{O}}\right), P_\theta\left(X_{\mathbf{O}}\right)\right). \tag{5}
$$

Here we note that we have $\overset{(1)}{=}$ because the expectation is preserved through a deterministic push-forward map.

Finally, combining (4) and (5), we reach the conclusion. □

It is worth noting that according to Theorem 1, we need to solve the following OP:

$$
\inf_{\left[\phi_i \in \mathfrak{C}(X_i)\right]_{i \in \mathbf{O}}} \mathbb{E}_{X_{\mathbf{O}} \sim P_d(X_{\mathbf{O}}), \mathrm{PA}_{X_{\mathbf{O}}} \sim \phi(X_{\mathbf{O}})} \left[ c(X_{\mathbf{O}}, \psi_\theta(\mathrm{PA}_{X_{\mathbf{O}}})) \right], \tag{6}
$$

where $\mathfrak{C}(X_i) = \{\phi_i : \phi_i \# P_d(X_i) = P_\theta(\mathrm{PA}_{X_i})\}, \forall i \in \mathbf{O}$.

If we make some further assumptions including: (i) the family model distributions $P_\theta, \theta \in \Theta$ induced by the graphical model is sufficiently rich to contain the data distribution, meaning that there exist $\theta^* \in \Theta$ such that $P_{\theta^*}(X_{\mathbf{O}}) = P_d(X_{\mathbf{O}})$ and (ii) the family of backward maps $\phi_i, i \in \mathbf{O}$ has infinite capacity (i.e., they include all measure functions), the infimum really peaks 0 at an optimal backward maps $\phi_i^*, i \in \mathbf{O}$. We thus can replace the infimum by a minimization as

$$
\min_{\left[\phi_i \in \mathfrak{C}(X_i)\right]_{i \in \mathbf{O}}} \mathbb{E}_{X_{\mathbf{O}} \sim P_d(X_{\mathbf{O}}), \mathrm{PA}_{X_{\mathbf{O}}} \sim \phi(X_{\mathbf{O}})} \left[ c(X_{\mathbf{O}}, \psi_\theta(\mathrm{PA}_{X_{\mathbf{O}}})) \right]. \tag{7}
$$

To make the OP in (7) tractable for training, we do relaxation as

$$
\min_\phi \left\{ \mathbb{E}_{X_{\mathbf{O}} \sim P_d(X_{\mathbf{O}}), \mathrm{PA}_{X_{\mathbf{O}}} \sim \phi(X_{\mathbf{O}})} \left[ c(X_{\mathbf{O}}, \psi_\theta(\mathrm{PA}_{X_{\mathbf{O}}})) \right] + \eta D\left(P_\phi, P_\theta\left(\mathrm{PA}_{X_{\mathbf{O}}}\right)\right) \right\}, \tag{8}
$$

where $\eta > 0$, $P_\phi$ is the distribution induced by the backward maps, and $D$ represents a general divergence. Here we note that $D\left(P_\phi, P_\theta\left(\mathrm{PA}_{X_{\mathbf{O}}}\right)\right)$ can be decomposed into

$$
D\left(P_\phi, P_\theta\left(\mathrm{PA}_{X_{\mathbf{O}}}\right)\right) = \sum_{i \in \mathbf{O}} D_i\left(P_{\phi_i}, P_\theta\left(\mathrm{PA}_{X_{\mathbf{i}}}\right)\right),
$$

which is the sum of the divergences between the specific backward map distributions and their corresponding model distributions on the parent nodes (i.e., $P_{\phi_i} = \phi_i \# P_d(X_i)$). Additionally, in practice, using the WS distance for $D_i$ leads to the following OP

$$
\min_\phi \left\{ \mathbb{E}_{X_{\mathbf{O}} \sim P_d(X_{\mathbf{O}}), \mathrm{PA}_{X_{\mathbf{O}}} \sim \phi(X_{\mathbf{O}})} \left[ c(X_{\mathbf{O}}, \psi_\theta(\mathrm{PA}_{X_{\mathbf{O}}})) \right] + \eta \sum_{i \in \mathbf{O}} W_{c_i}\left(P_{\phi_i}, P_\theta\left(\mathrm{PA}_{X_i}\right)\right) \right\}. \tag{9}
$$

The following theorem characterizes the ability to search the optimal solutions for the OPs in (7), (8), and (9).

**Theorem 2.** *Assume that the family model distributions $P_\theta, \theta \in \Theta$ induced by the graphical model is sufficiently rich to contain the data distribution, meaning that there exist $\theta^* \in \Theta$ such that $P_{\theta^*}(X_{\mathbf{O}}) =*

$P_d(X_\mathbf{O})$ *and the family of backward maps* $\phi_i, i \in \mathbf{O}$ *has infinite capacity (i.e., they include all measure functions). The OPs in* (7), (8), *and* (9) *are equivalent and can obtain the common optimal solution.*

*Proof.* Let $\theta^* \in \Theta$ be the optimal solution such that $P_{\theta^*}(X_\mathbf{O}) = P_d(X_\mathbf{O})$ and $W_c\left(P_d\left(X_\mathbf{O}\right), P_{\theta^*}\left(X_\mathbf{O}\right)\right) = 0$. Let $\Gamma^* \in \mathcal{P}(P_d(X_\mathbf{O}), P_\theta(X_\mathbf{O}))$ be the optimal joint distribution over $P_d(X_\mathbf{O})$ and $P_\theta(X_\mathbf{O})$ of the corresponding Wasserstein distance, meaning that if $(X_\mathbf{O}, \tilde{X}_\mathbf{O}) \sim \Gamma^*$ then $X_\mathbf{O} = \tilde{X}_\mathbf{O}$. Using the gluing lemma as in the previous theorem, there exists a joint distribution $\alpha^*$ over $A \times B \times C$ such that $\alpha_{AC}^* = (\pi_A, \pi_C)\#\alpha^* = \Gamma^*$ and $\alpha_{BC}^* = (\pi_B, \pi_C)\#\alpha^* = \beta^*$ where $\beta^* = (id, \psi_\theta)\#P_\theta^*([\mathrm{PA}_{X_i}]_{i \in \mathbf{O}})$ is a deterministic coupling or joint distribution over $P_\theta([\mathrm{PA}_{X_i}]_{i \in \mathbf{O}})$ and $P_\theta^*(X_\mathbf{O})$. This follows that $\alpha^*$ consists of the sample $(X_\mathbf{O}, \mathrm{PA}_{X_\mathbf{O}}, X_\mathbf{O})$ where $\psi_{\theta^*}(\mathrm{PA}_{X_\mathbf{O}}) = X_\mathbf{O}$ with $X_\mathbf{O} \sim P_d(X_\mathbf{O}) = P_\theta^*(X_\mathbf{O})$.

Let us denote $\gamma^* = (\pi_A, \pi_B)\#\alpha^*$ as a joint distribution over $P_d(X_\mathbf{O})$ and $P_\theta^*([\mathrm{PA}_{X_i}]_{i \in \mathbf{O}})$. Let $\gamma_i^*, i \in \mathbf{O}$ as the restriction of $\gamma^*$ over $P_d(X_i)$ and $P_\theta^*(\mathrm{PA}_{X_i})$. Let $\phi_i^*, i \in \mathbf{O}$ be the functions in the family of the backward functions that can well-approximate $\gamma_i^*, i \in \mathbf{O}$ (i.e., $\phi_i^* = \gamma_i^*, i \in \mathbf{O}$). For any $X_\mathbf{O} \sim P_d(X_\mathbf{O})$, we have for all $i \in \mathbf{O}$, $\mathrm{PA}_{X_i} = \phi_i^*(X_i)$ and $\psi_{\theta^*}(\mathrm{PA}_{X_i}) = X_i$. These imply that (i) $\mathbb{E}_{X_\mathbf{O} \sim P_d(X_\mathbf{O}), \mathrm{PA}_{X_\mathbf{O}} \sim \phi^*(X_\mathbf{O})}\left[c\left(X_\mathbf{O}, \psi_{\theta^*}(\mathrm{PA}_{X_\mathbf{O}})\right)\right] = 0$ and (ii) $P_{\phi_i^*} = P_{\theta^*}(\mathrm{PA}_{X_i}), \forall i \in \mathbf{O}$, which further indicate that the OPs in (7), (8), and (9) are minimized at 0 with the common optimal solution $\phi^*$ and $\theta^*$. □

## B   RELATED WORK

Optimization of the ELBO encounter many practical challenges. For vanilla EM, the algorithm only works if the true posterior density can be computed exactly. Furthermore, EM is originally a batch algorithm, thereby converging slowly on large datasets (Liang & Klein, 2009). Subsequently, researchers have tried exploring other methods for scalability, including attempts to combine EM with approximate inference (Wei & Tanner, 1990; Neal & Hinton, 1998; Delyon et al., 1999; Beal & Ghahramani, 2006; Cappé & Moulines, 2009; Liang & Klein, 2009; Neath et al., 2013).

When exact inference is infeasible, a variational approximation is the go-to solution. Along this line, research efforts have concentrated on ensuring tractability of the ELBO via the mean-field assumption (Bishop & Nasrabadi, 2006) and its relaxation known as structured mean field (Saul & Jordan, 1995). Scalability has been one of the main challenges facing the early VI formulations since it is a batch algorithm. This has triggered the development of stochastic variational inference(SVI, Hoffman et al., 2013; Hoffman & Blei, 2015; Foti et al., 2014; Johnson & Willsky, 2014; Anandkumar et al., 2012; 2014) which applies stochastic optimization to solve VI objectives. Another line of work is collapsed VI that explicitly integrates out certain model parameters or latent variables in an analytic manner (Hensman et al., 2012; King & Lawrence, 2006; Teh et al., 2006; Lázaro-Gredilla et al., 2012). Without a closed form, one could resort to Markov chain Monte Carlo (Gelfand & Smith, 1990; Gilks et al., 1995; Hammersley, 2013), which however tends to be slow. More accurate variational posteriors also exist, namely, through hierarchical variational models (Ranganath et al., 2016), implicit posteriors (Titsias & Ruiz, 2019; Yin & Zhou, 2018; Molchanov et al., 2019; Titsias & Ruiz, 2019), normalizing flows (Kingma et al., 2016), or copula distribution (Tran et al., 2015). To avoid computing the ELBO analytically, one can obtain an unbiased gradient estimator using Monte Carlo and re-parameterization tricks (Ranganath et al., 2014; Xu et al., 2019). As mentioned in the introduction, an excellent summary of these approaches is discussed in (Ambrogioni et al., 2021, §6). Extensions of VI to other divergence measures than KL divergence e.g., $\alpha-$divergence or $f-$divergence, also exist (Li & Turner, 2016; Hernandez-Lobato et al., 2016; Wan et al., 2020). In the causal inference literature, a related direction is to learn both the graphical structure and parameters of the corresponding structural equation model (Yu et al., 2019; Geffner et al., 2022). These frameworks are often limited to additive noise models while assuming no latent confounders.

## C    TRAINING ALGORITHMS

Algorithm 1 provides the pseudo-code for OTP-DAG learning procedure. The simplicity of the learning process is evident. Figure 5a visualizes our backward-forward algorithm in the empirical setting, where learning the backward functions for the endogenous variables only is sufficient for estimation. Regardless of the complexity of the graphical structure, a single learning procedure is applied. The first step is to identify the observed nodes and their parent nodes; then, for each parent-child pair, define the appropriate backward map and reparameterize the model distribution into a set of deterministic forward maps parameterized by $\theta$ (i.e., model parameters to be learned). Finally, one only needs to plug in the suitable cost function and divergence measure, and follow the backward-forward procedure to learn $\theta$ via stochastic gradient descent.

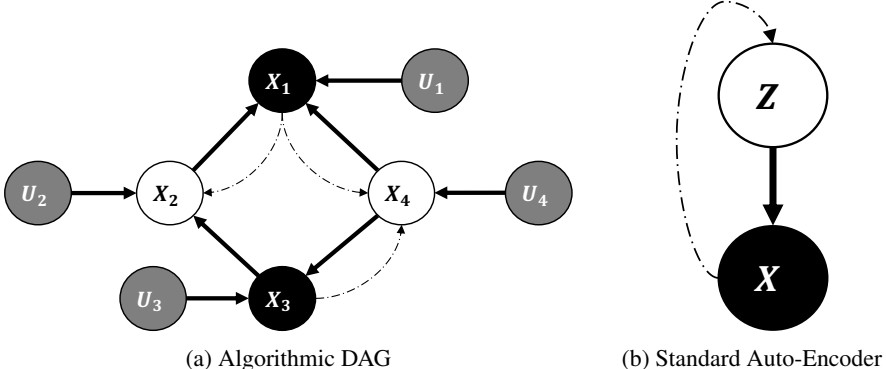

(a) Algorithmic DAG                        (b) Standard Auto-Encoder

Figure 5

## D    CONNECTION WITH AUTO-ENCODERS

In this section, we elaborate on our discussion in Section 3. Figure 5b sheds light on an interesting connection of our method with auto-encoding models. Considering a graphical model of only two nodes: the observed node $X$ and its latent parent $Z$, we define a backward map $\phi$ over $X$ such that $\phi \# P_d(X) = P_\theta(Z)$ where $P_\theta(Z)$ is the prior over $Z$. The backward map can be viewed as a (stochastic) encoder approximating the poster $P_\theta(Z)$ with $P_\phi(Z|X)$. OTP-DAG now reduces to Wasserstein auto-encoder WAE (Tolstikhin et al., 2017), where the forward mapping $\psi$ plays the role of the decoder. OTP-DAG therefore serves as a generalization of WAE for learning a more complex structure where there is the interplay of more parameters and hidden variables.

In this simplistic case, our training procedure is precisely as follows:

1. Draw $X \sim P_d(X)$.
2. Draw $Z \sim \phi(X)$.
3. Draw $\widetilde{X} \sim P_\theta(X|Z)$.
4. Evaluate the costs according to Eq. 3 and update $\theta$.

Our cost function explicitly minimizes two terms: (1) the push-forward divergence $D[P_\phi(Z|X), P_\theta(Z)]$ where $D$ is an arbitrary divergence (we use the WS distance for $D$ in our experiments), and (2) the reconstruction loss between $X$ and $\widetilde{X}$.

**Posterior Collapse**    Relaxing the push-forward constraint into the divergence term means the backward $\phi$ is forced to mimic the prior, which may lead to a situation similar to posterior collapse notoriously occurring to VAE. We here detail why VAE is prone to this issue and how the OT-based objective mitigates it.

---

**Algorithm 1:** OTP-DAG Algorithm

---

**Input:** Directed graph $\mathbf{G}$ with observed nodes $\mathbf{O}$, noise distribution $P(U)$, stochastic backward maps $\phi = \{\phi_i(X_i)\}_{i \in \mathbf{O}}$, regularization coefficient $\eta$, reconstruction cost function $c$, and push-forward divergence measure $D$.

**Output:** Point estimate $\theta$.

Re-parameterize $P_\theta$ into a set of deterministic mappings $\psi_\theta = \{\psi_{\theta_i}\}_{i \in \mathbf{O}}$ where $X_i = \psi_{\theta_i}(\mathrm{PA}_{X_i}, U_i)$ and $U_i \sim P(U)$.

Initialize the parameters of the forward $\psi_\theta$ and backward $\phi$ mapping functions.

**while** *not converged* **do**

    **for** $i \in \mathbf{O}$ **do**

        Sample batch $X_i^B = \{x_i^1, ..., x_i^B\}$;

        Sample $\widetilde{\mathrm{PA}}_{X_i^B}$ from $\phi_i(X_i^B)$;

        Sampling $U_i$ from the prior $P(U)$;

        Evaluate $\widetilde{X}_i^B = \psi_{\theta_i}(\widetilde{\mathrm{PA}}_{X_i^B}, U_i)$.

    **end**

    Update $\theta$ by descending

$$\frac{1}{B} \sum_{b=1}^{B} \sum_{i \in \mathbf{O}} c\big(x_i^b, \widetilde{x}_i^b\big) + \eta \, D\big[P_{\phi_i}(\mathrm{PA}_{X_i^B}|X_i^B), P_\theta(\mathrm{PA}_{X_i^B})\big]$$

**end**

---

Intuitively, our learning dynamic seeks to ensures $\phi_\# P_d(X) = P_\phi(Z|X) = P_\theta(Z)$ so that $Z \sim P_\phi(Z|X)$ follows the prior distribution $P_\theta(Z)$. However, such samples $Z \sim \phi(X)$ cannot ignore information in the input $X$ due to minimizing the reconstruction term, that is, we need $X \sim P_d(X)$ and $\widetilde{X} \sim P_\theta(X \mid Z)$ to close and $\widetilde{X}$ should follow the data distribution $P_d(X)$.

**1. The push-forward divergence**

While the objectives of OTP-DAG/WAE and VAE entail the prior matching term. the two formulations are different in nature.

Let $Q$ denote the set of variational distributions. The VAE objective can be written as

$$\inf_{\phi(Z|X) \in Q} \mathbb{E}_{X \sim P(X)}[D_{\mathrm{KL}}(\phi(Z|X), P_\theta(Z))] - \mathbb{E}_{Z \sim \phi(Z|X)}[\log P_\theta(X|Z)]. \tag{10}$$

By minimizing the above KL divergence term, VAE basically tries to match the prior $P(Z)$ for all different examples drawn from $P_d(X)$. Under the VAE objective, it is thus easier for $\phi$ to collapse into a distribution independent of $P_d(X)$, where specifically latent codes are close to each other and reconstructed samples are concentrated around only few values.

For OTP-DAG/WAE, the regularizer in fact penalizes the discrepancy between $P_\theta(Z)$ and $P_\phi := \mathbb{E}_{P(X)}[\phi(X)]$, which can be optimized using GAN-based, MMD-based or Wasserstein distance. The latent codes of different examples $X \sim P_d(X)$ can lie far away from each other, which allows the model to maintain the dependency between the latent codes and the input. Therefore, it is more difficult for $\phi$ to mimic the prior and trivially satisfy the push-forward constraint. We refer readers to Tolstikhin et al. (2017) for extensive empirical evidence.

**2. The reconstruction loss**

At some point of training, there is still a possibility to land at $\phi$ that yields samples $Z$ independent of input $X$. If this occurs, $\phi \# \delta x_c^{(1)} = \phi \# \delta x_c^{(2)} = P(Z)$ for any points $x_c^{(1)}, x_c^{(2)} \sim P_d(X)$. This means $\text{supp}(\phi(X^1)) = \text{supp}(\phi(X^2)) = \text{supp}(P_\theta(Z))$, so it would result in a very large reconstruction loss because it requires to map $\text{supp}(P(Z))$ to various $X^1$ and $X^2$. Thus our reconstruction term would heavily penalizes this. In other words, this term explicitly encourages the model to search for $\theta$ that reconstruct better, thus preventing the model from converging to the backward $\phi$ that produces sub-optimal ancestral samples.

Meanwhile, for VAE, if the family $Q$ contains all possible conditional distribution $\phi(Z|X)$, its objective is essentially to maximize the marginal log-likelihood $\mathbb{E}_{P(X)}[\log P_\theta(X)]$, or minimize the KL divergence $\text{KL}(P_d, P_\theta)$. It is shown in Dai et al. (2020) that under posterior collapse, VAE produces poor reconstructions yet the loss can still decrease i.e achieve low negative log-likelihood scores and still able to assign high-probability to the training data.

In summary, it is such construction and optimization of the backward that prevents OTP-DAG from posterior collapse situation. We here search for $\phi$ within a family of measurable functions and in practice approximate it with deep neural networks. It comes down to empirical decisions to select the architecture sufficiently expressive to each application.

**Additional Experiment**   We here study the capability of recovering the true parameters of semi-amortized VAE (SA-VAE) in comparison with our OTP-DAG. We borrow the setting in Section 4.1 of the paper Kim et al. (2018). We create a synthetic dataset from a generative model of discrete sequences according to an oracle generative process as follows:

$$\mathbf{z} \sim \mathcal{N}(0, \mathbf{I})$$
$$\mathbf{h}_t = \text{LSTM}(\mathbf{h}_{t-1}, \mathbf{x}_t)$$
$$\mathbf{x}_{t+1} \sim \text{softmax}(\text{MLP}([\mathbf{h}_t, \mathbf{z}]))$$

The architecture is a 1-layer LSTM with 50 hidden units where the input embedding is also 50-dimensional. The initial hidden/cell states are set to zero. We generate for $T = 5$ time steps for each example i.e., each input $x$ is a $5-$dimensional vector. The MLP consists of a single affine transformation to project out to the vocabulary space of size 100. The latent variable $z$ is a 50-dimensional vector.

We here assume the architecture of the oracle is known and the task is simply to learn the parameters from $10,000$ examples. Table 4 reports how well the estimated parameters approximate the ground-truth in terms of mean absolute error (MDE) and mean squared error (MSE). NLL reports the negative log-likelihood loss of $50,000$ reconstructed samples from the generative model given the learned latent representations. We also report the performance of a randomly initialized decoder to highlight the effect of learning. This empirical evidence again substantiates our competitiveness with amortization inference methods.

Table 4: Fidelity of estimated parameters of the oracle generative model.

| Model | MDE $\downarrow$ | MSE $\downarrow$ | NLL $\downarrow$ |
|---|---|---|---|
| **OTP-DAG** (Ours) | $0.885 \pm 0.000$ | $\mathbf{1.790 \pm 0.000}$ | $\mathbf{-0.951 \pm 0.002}$ |
| **SA-VAE** | $\mathbf{0.878 \pm 0.000}$ | $\mathbf{1.790 \pm 0.000}$ | $-0.949 \pm 0.001$ |
| **VAE** | $0.890 \pm 0.000$ | $1.829 \pm 0.000$ | $-0.468 \pm 0.001$ |
| **Random** | $1.192 \pm 0.040$ | $3.772 \pm 0.235$ | $-0.020 \pm 0.007$ |

# E   EXPERIMENTAL SETUP

In the following, we explain how OTP-DAG algorithm is implemented in practical applications, including how to reparameterize the model distribution, to design the backward mapping and to define the optimization objective. We also here provide the training configurations for our method and the baselines. All models

are run on $4$ RTX 6000 GPU cores using Adam optimizer with a fixed learning rate of $1e - 3$. Our code is anonymously published at https://anonymous.4open.science/r/OTP-7944/.

### E.1 Latent Dirichlet Allocation

For completeness, let us recap the model generative process. We consider a corpus $\mathcal{D}$ of $M$ independent documents where each document is a sequence of $N$ words denoted by $W_{1:N} = (W_1, W_2, \cdots, W_N)$. Documents are represented as random mixtures over $K$ latent topics, each of which is characterized by a distribution over words. Let $V$ be the size of a vocabulary indexed by $\{1, \cdots, V\}$. Latent Dirichlet Allocation (LDA) Blei et al. (2003) dictates the following generative process for every document in the corpus:

1. Choose $\theta \sim \text{Dir}(\alpha)$,
2. Choose $\gamma_k \sim \text{Dir}(\beta)$ where $k \in \{1, \cdots, K\}$,
3. For each of the word positions $n \in \{1, \cdots, N\}$,
   - Choose a topic $z_n \sim \text{Multi-Nominal}(\theta)$,
   - Choose a word $w_n \sim \text{Multi-Nominal}(z_n, \gamma_k)$,

where $\text{Dir}(.)$ is a Dirichlet distribution, $\alpha < 1$ and $\beta$ is typically sparse. $\theta$ is a $K-$dimensional vector that lies in the $(K-1)-$simplex and $\gamma_k$ is a $V-$dimensional vector represents the word distribution corresponding to topic $k$. Throughout the experiments, $K$ is fixed at $10$.

**Parameter Estimation.** We consider the topic-word distribution $\gamma$ as a fixed quantity to be estimated. $\gamma$ is a $K \times V$ matrix where $\gamma_{kn} := P(W_n = 1 | Z_n = 1)$. The learnable parameters therefore consist of $\gamma$ and $\alpha$. An input document is represented with a $N \times V$ matrix where a word $W_i$ is represented with a one-hot $V-$vector such that the value at the index $i$ in the vocabulary is $1$ and $0$ otherwise. Given $\gamma \in [0,1]^{K \times V}$ and a selected topic $k$, the deterministic forward mapping to generate a document $W$ is defined as

$$W_{1:N} = \psi(Z) = \text{Cat-Concrete}\big(\text{softmax}(Z'\gamma)\big),$$

where $Z \in \{0,1\}^K$ is in the one-hot representation (i.e., $Z^k = 1$ if state $k$ is the selected and $0$ otherwise) and $Z'$ is its transpose. By applying the Gumbel-Softmax trick Jang et al. (2016); Maddison et al. (2016), we re-parameterize the Categorical distribution into a function Cat-Concrete$(.)$ that takes the categorical probability vector (i.e., sum of all elements equals $1$) and output a relaxed probability vector. To be more specific, given a categorical variable of $K$ categories with probabilities $[p_1, p_2, ..., p_K]$, for every the Cat-Concrete$(.)$ function is defined on each $p_k$ as

$$\text{Cat-Concrete}(p_k) = \frac{\exp\big\{(\log p_k + G_k)/\tau\big\}}{\sum_{k=1}^K \exp\big\{(\log p_k + G_k)/\tau\big\}},$$

with temperature $\tau$, random noises $G_k$ independently drawn from Gumbel distribution $G_t = -\log(-\log u_t)$, $u_t \sim \text{Uniform}(0,1)$.

We next define a backward map that outputs for a document a distribution over $K$ topics as follows

$$\phi(W_{1:N}) = \text{Cat}(Z).$$

Given observations $W_{1:N}$, our learning procedure begins by sampling $\widetilde{Z} \sim P_\phi(Z|W_{1:N})$ and pass $\widetilde{Z}$ through the generative process given by $\psi$ to obtain the reconstruction. Notice here that we have a prior constraint over the distribution of $\theta$ i.e., $\theta$ follows a Dirichlet distribution parameterized by $\alpha$. This translates to a push forward constraint in order to optimize for $\alpha$. To facilitate differentiable training, we use softmax Laplace approximation (MacKay, 1998; Srivastava & Sutton, 2017) to approximate a Dirichlet distribution with a softmax Gaussian distribution. The relation between $\alpha$ and the Gaussian parameters $(\mu_k, \Sigma_k)$ w.r.t a category $k$ where $\Sigma_k$ is a diagonal matrix is given as

$$\mu_k(\alpha) = \log \alpha_k - \frac{1}{K} \sum_{i=1}^{K} \log \alpha_i, \quad \Sigma_k(\alpha) = \frac{1}{\alpha_k}\left(1 - \frac{2}{K}\right) + \frac{1}{K^2} \sum_{i=1}^{K} \frac{1}{\alpha_i}. \tag{11}$$

Let us denote $P_\alpha := \mathcal{N}\big(\mu(\alpha), \Sigma(\alpha)\big) \approx \mathrm{Dir}(\alpha)$ with $\mu = [\mu_k]_{k=1}^{K}$ and $\Sigma = [\Sigma_k]_{k=1}^{K}$ defined as above. We learn $\alpha, \gamma$ by minimizing the following optimization objective

$$\mathbb{E}_{W_{1:N}, \widetilde{Z}}\left[c\big(W_{1:N}, \psi(\widetilde{Z})\big) + \eta\, D_{\mathrm{WS}}\big[P_\phi(Z|W_{1:N}), \theta\big]\right],$$

where $W_{1:N} \sim \mathcal{D}, \widetilde{Z} \sim P_\phi(Z|W_{1:N}), \theta \sim P_\alpha$, $c$ is cross-entropy loss function and $D_{\mathrm{WS}}$ is exact Wasserstein distance[3]. The sampling process $\theta \sim P_\alpha$ is also relaxed using standard Gaussian reparameterization trick whereby $\theta = \mu(\alpha) + u\Sigma(\alpha)$ with $u \sim \mathcal{N}(0,1)$.

**Remark.** Our framework in fact learns both $\alpha$ and $\gamma$ at the same time. Our estimates for $\alpha$ (averaged over $K$) are nearly $100\%$ faithful at $0.10, 0.049, 0.033$ (recall that the ground-truth $\alpha$ is uniform over $K$ where $K = 10, 20, 30$ respectively). Figure 6 shows the convergence behavior of OTP-DAG during training where our model converges to the ground-truth patterns relatively quickly.

Figures 7 and 8 additionally present the topic distributions of each method for the second and third synthetic sets. We use horizontal and vertical patterns in different colors to distinguish topics from one another. Red circles indicate erroneous patterns. Note that these configurations are increasingly more complex, so it may require more training time for all methods to achieve better performance. Regardless, it is seen that although our method may exhibit some inconsistencies in recovering accurate word distributions for each topic, these discrepancies are comparatively less pronounced when compared to the baseline methods. This observation indicates a certain level of robustness in our approach.

**Topic Inference.** In this experiment, we apply OTP-DAG on real-world topic modeling tasks. We here revert to the original generative process where the topic-word distribution follows a Dirichlet distribution parameterized by the concentration parameters $\beta$, instead of having $\gamma$ as a fixed quantity. In this case, $\beta$ is initialized as a matrix of real values i.e., $\beta \in \mathbb{R}^{K \times V}$ representing the log concentration values. The forward process is given as

$$W_{1:N} = \psi(Z) = \text{Cat-Concrete}\big(\text{softmax}(Z'\gamma)\big),$$

where $\gamma_k = \mu_k\big(\exp(\beta_k)\big) + u_k \Sigma_k\big(\exp(\beta_k)\big)$ and $u_k \sim \mathcal{N}(0,1)$ is a Gaussian noise. This is realized by using softmax Gaussian trick as in Eq. (11), then applying standard Gaussian reparameterization trick. The optimization procedure follows the previous application.

**Remark.** The datasets and the implementation of Prod LDA are provided in the OCTIS library at https://github.com/MIND-Lab/OCTIS. We also use OCTIS to standardize evaluation for all models on the topic inference task. Note that the computation of topic coherence score using normalized pointwise mutual information in OCTIS is different than in Srivastava & Sutton (2017).

For every topic $k$, we select top 10 most related words according to $\gamma_k$ to represent it. This task assesses the quality of the inferred topics on real-world datasets. Topic quality is evaluated via the diversity and coherence of the selected words. Diversity refers to the proportion of unique words, whereas Coherence is measured with normalized pointwise mutual information (Aletras & Stevenson, 2013), reflecting the extent to which the words in a topic are associated with a common theme.

Tables 5 and 6 respectively present the quantitative and qualitative results. Due to the trade-off between Coherence and Diversity, model performance should be judged comprehensively. In this regard, OTP-DAG scores competitively high on both metrics and consistently across different settings.

---

[3]https://pythonot.github.io/index.html

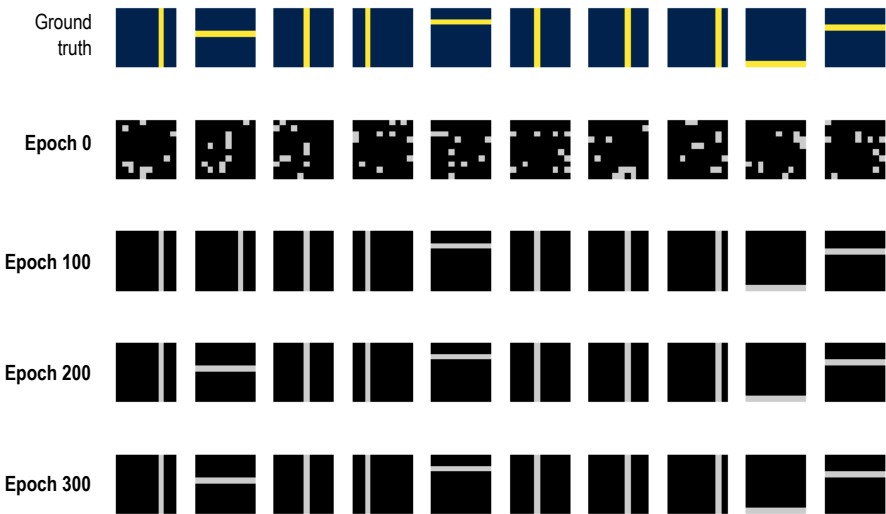

Figure 6: Converging patterns of 10 random topics from our OTP-DAG after 100, 200, 300 iterations.

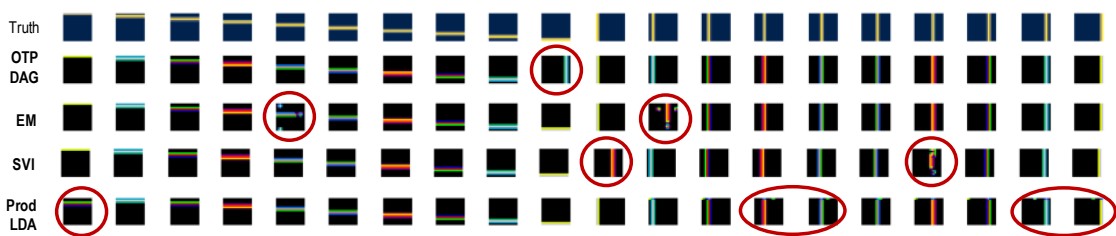

Figure 7: Topic-word distributions inferred by OTP-DAG from the second set of synthetic data after 300 training epochs.

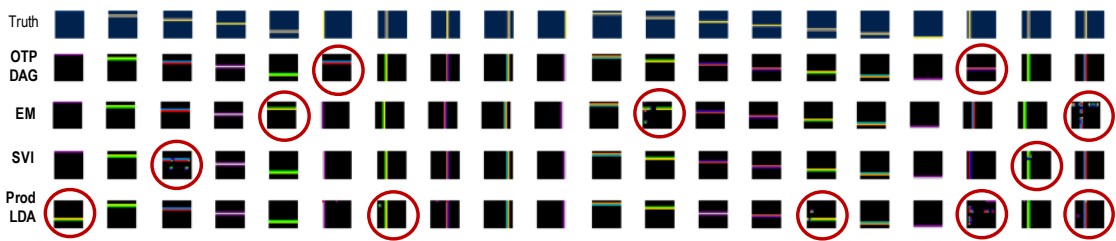

Figure 8: Topic-word distributions inferred by OTP-DAG from the third set of synthetic data after 300 training epochs.

**Training Configuration.** The underlying architecture of the backward maps consists of an LSTM and one or more linear layers. We train all models for 300 and 1,000 epochs with batch size of 50 respectively for the 2 applications. We also set $\tau = 1.0, 2.0$ and $\eta = 1e - 4, 1e - 1$ respectively.

Table 5: Coherence and Diversity of the inferred topics for the 3 real-world datasets ($K = 10$)

| Metric (%) ↑ | OTP-DAG (Ours) | Batch EM | SVI | Prod LDA |
|---|---|---|---|---|
| Coherence | **10.46 ± 0.13** | 6.71 ± 0.16 | 5.90 ± 0.51 | 4.78 ± 2.64 |
| Diversity | **93.33 ± 4.62** | 72.33 ± 1.15 | 85.33 ± 5.51 | 92.67 ± 4.51 |
| Coherence | **9.03 ± 1.00** | 8.67 ± 0.62 | 7.84 ± 0.49 | 2.17 ± 2.36 |
| Diversity | 84.33 ± 2.08 | 86.00 ± 1.00 | **92.33 ± 2.31** | 87.67 ± 3.79 |
| Coherence | **4.63 ± 0.22** | 4.52 ± 0.53 | 1.47 ± 0.39 | 2.91 ± 1.70 |
| Diversity | **98.73 ± 1.15** | 81.33 ± 1.15 | 92.67 ± 2.52 | 98.67 ± 1.53 |

Table 6: Topics inferred for 3 real-world datasets.

| 20 News Group | |
|---|---|
| Topic 1 | *car, bike, front, engine, mile, ride, drive, owner, road, buy* |
| Topic 2 | *game, play, team, player, season, fan, win, hit, year, score* |
| Topic 3 | *government, public, key, clipper, security, encryption, law, agency, private, technology* |
| Topic 4 | *religion, christian, belief, church, argument, faith, truth, evidence, human, life* |
| Topic 5 | *window, file, program, software, application, graphic, display, user, screen, format* |
| Topic 6 | *mail, sell, price, email, interested, sale, offer, reply, info, send* |
| Topic 7 | *card, drive, disk, monitor, chip, video, speed, memory, system, board* |
| Topic 8 | *kill, gun, government, war, child, law, country, crime, weapon, death* |
| Topic 9 | *make, time, good, people, find, thing, give, work, problem, call* |
| Topic 10 | *fire, day, hour, night, burn, doctor, woman, water, food, body* |

| BBC News | |
|---|---|
| Topic 1 | *rise, growth, market, fall, month, high, economy, expect, economic, price* |
| Topic 2 | *win, play, game, player, good, back, match, team, final, side* |
| Topic 3 | *user, firm, website, computer, net, information, software, internet, system, technology* |
| Topic 4 | *technology, market, digital, high, video, player, company, launch, mobile, phone* |
| Topic 5 | *election, government, party, labour, leader, plan, story, general, public, minister* |
| Topic 6 | *film, include, star, award, good, win, show, top, play, actor* |
| Topic 7 | *charge, case, face, claim, court, ban, lawyer, guilty, drug, trial* |
| Topic 8 | *thing, work, part, life, find, idea, give, world, real, good* |
| Topic 9 | *company, firm, deal, share, buy, business, market, executive, pay, group* |
| Topic 10 | *government, law, issue, spokesman, call, minister, public, give, rule, plan* |

| DBLP | |
|---|---|
| Topic 1 | *learning, algorithm, time, rule, temporal, logic, framework, real, performance, function* |
| Topic 2 | *efficient, classification, semantic, multiple, constraint, optimization, probabilistic, domain, process, inference* |
| Topic 3 | *search, structure, pattern, large, language, web, problem, representation, support, machine* |
| Topic 4 | *object, detection, application, information, method, estimation, multi, dynamic, tree, motion* |
| Topic 5 | *system, database, query, knowledge, processing, management, orient, relational, expert, transaction* |
| Topic 6 | *model, markov, mixture, variable, gaussian, topic, hide, latent, graphical, appearance* |
| Topic 7 | *network, approach, recognition, neural, face, bayesian, belief, speech, sensor, artificial* |
| Topic 8 | *base, video, content, code, coding, scalable, rate, streaming, frame, distortion* |
| Topic 9 | *datum, analysis, feature, mining, cluster, selection, high, stream, dimensional, component* |
| Topic 10 | *image, learn, segmentation, retrieval, color, wavelet, region, texture, transform, compression* |

### E.2  HIDDEN MARKOV MODELS

We here attempt to learn a Poisson hidden Markov model underlying a data stream. Given a time series $\mathcal{D}$ of $T$ steps, the task is to segment the data stream into $K$ different states, each of which is associated with a Poisson observation model with rate $\lambda_k$. The observation at each step $t$ is given as

$$P(X_t|Z_t = k) = \text{Poi}(X_t|\lambda_k), \quad \text{for } k = 1, \cdots, K.$$

The Markov chain stays in the current state with probability $p$ and otherwise transitions to one of the other $K - 1$ states uniformly at random. The transition distribution is given as

$$Z_1 \sim \text{Cat}\left(\left\{\frac{1}{4}, \frac{1}{4}, \frac{1}{4}, \frac{1}{4}\right\}\right), \quad Z_t|Z_{t-1} \sim \text{Cat}\left(\left\{ \begin{array}{ll} p & \text{if } Z_t = Z_{t-1} \\ \frac{1-p}{4-1} & \text{otherwise} \end{array} \right\}\right)$$

Let $P(Z_1)$ and $P(Z_t|Z_{t-1})$ respectively denote these prior transition distributions. We first apply Gaussian reparameterization on each Poisson distribution, giving rise to a deterministic forward mapping

$$X_t = \psi_t(Z_t) = Z'_t \exp(\lambda) + u_t \sqrt{Z_t \exp(\lambda)},$$

where $\lambda \in \mathbb{R}^K$ is the learnable parameter vector representing log rates, $u_k \sim \mathcal{N}(0, 1)$ is a Gaussian noise, $Z_t \in \{0, 1\}^K$ is in the one-hot representation and $Z'_t$ is its transpose. We define a global backward map $\phi$ that outputs the distributions for individual $Z_t$ as $\phi(X_t) := \text{Cat}(Z_t)$.

The first term in the optimization object is the reconstruction error given by a cost function $c$. The push forward constraint ensures the backward probabilities for the state variables align with the prior transition distributions. Putting everything together, we learn $\lambda_{1:K}$ by minimizing the following empirical objective

$$\mathbb{E}_{X_{1:T}, \widetilde{Z}_{1:T}} \left[ c\big(X_{1:T}, \psi(\widetilde{Z}_{1:T})\big) + \eta \, D_{\text{WS}}\big[P_\phi(Z_1|X_1), P(Z_1)\big] + \eta \sum_{t=2}^{T} D_{\text{WS}}\big[P_\phi(Z_t|X_t), P(Z_t|Z_{t-1})\big] \right],$$

where $X_{1:T} \sim \mathcal{D}$, $\widetilde{Z}_{1:T} \sim P_\phi(Z_{1:T}|X_{1:T})$ and $\psi = [\psi_t]_{t=1}^T$.

In this application, $T = 200$ and smooth $L_1$ loss (Girshick, 2015) is chosen as the cost function. $D_{\text{WS}}$ is exact Wasserstein distance with KL divergence as the ground cost. We compute MAP estimates of the Poisson rates using stochastic gradient descent, using a $\log - \text{Normal}(5, 5)$ prior for $p(\lambda)$.

**Training Configuration.**  The underlying architecture of the backward map is a $3-$ layer fully connected perceptron. The Poisson HMM is trained for $20,000$ epochs with $\eta = 0.1$ and $\tau = 0.1$.

**Additional Experiment.**  As discussed in Section 4.2, $p = 0.05$ is evidently a poor choice of transition probability to fit the model. With $\eta = 0.1$, the regularization effect of the push-forward divergence is relatively weak. Thus, our model can still effectively converge to the true values.

However, if we increase the $\eta$ weight and strongly force the model to fit $p = 0.05$, the model performance degrades, in terms of both estimation and reconstruction quality. Table 7 provides empirical evidence for this claim, where we report OTP-DAG estimates and reconstruction losses at each $\eta$ value. It can be seen that the model poorly fits the data if strongly forced to fit the wrong prior probabilities.

### E.3  LEARNING DISCRETE REPRESENTATIONS

To understand vector quantized models, let us briefly review Quantization Variational Auto-Encoder (VQ-VAE) Van Den Oord et al. (2017). The practical setting of VQ-VAE in fact considers a $M-$dimensional discrete latent space $\mathcal{C}^M \in \mathbb{R}^{M \times D}$ that is the $M-$ary Cartesian power of $\mathcal{C}$ with $\mathcal{C} = \{c_k\}_{k=1}^K \in \mathbb{R}^{K \times D}$ i.e., $\mathcal{C}$ here is the set of learnable latent embedding vectors $c_k$. The latent variable $Z = [Z^m]_{m=1}^M$ is an $M-$component vector where each component $Z^m \in \mathcal{C}$. VQ-VAE is an encoder-decoder, in which the

Table 7: Estimates of $\lambda_{1:4}$ and mean absolute reconstruction error at $p = 0.05$ and various $\eta$ values.

| $\eta$ | $\lambda_1 = 12$ | $\lambda_2 = 87$ | $\lambda_3 = 60$ | $\lambda_4 = 33$ | MAE $\downarrow$ |
|---|---|---|---|---|---|
| 0.1 | 11.83 | 87.20 | 60.61 | 33.40 | 7.11 |
| 0.5 | 11.65 | 87.48 | 61.17 | 33.11 | 7.96 |
| 0.8 | 11.74 | 86.29 | 60.51 | 33.30 | 8.08 |
| 1.0 | 11.66 | 86.62 | 60.30 | 33.20 | 8.88 |
| 2.0 | 12.21 | 86.55 | 60.31 | 33.20 | 9.04 |
| 5.0 | 13.37 | 84.19 | 60.23 | 34.38 | 15.32 |

encoder $f_e : \mathcal{X} \mapsto \mathbb{R}^{M \times D}$ maps the input data $X$ to the latent representation $Z$ and the decoder $f_d :$ $\mathbb{R}^{M \times D} \mapsto \mathcal{X}$ reconstructs the input from the latent representation. However, different from standard VAE, the latent representation used for reconstruction is discrete, which is the projection of $Z$ onto $\mathcal{C}^M$ via the quantization process $Q$. Let $\bar{Z}$ denote the discrete representation. The quantization process is modeled as a deterministic categorical posterior distribution such that

$$\bar{Z}^m = Q(Z^m) = c_k,$$

where $k = \underset{k}{\operatorname{argmin}}\, d\big(Z^m, c_k\big)$, $Z^m = f_e^m(X)$ and $d$ is a metric on the latent space.

In our language, each vector $c_k$ can be viewed as the centroid representing each latent sub-space (or cluster). The quantization operation essentially searches for the closet cluster for every component latent representation $z^m$. VQ-VAE minimizes the following objective function:

$$\mathbb{E}_{x \sim \mathcal{D}}\left[d_x\big[f_d\big(Q(f_e(x))\big), x\big] + d_z\big[\mathbf{sg}\big(f_e(x)\big), \bar{z}\big] + \beta d_z\big[f_e(x), \mathbf{sg}(\bar{z})\big]\right],$$

where $\mathcal{D}$ is the empirical data, $\mathbf{sg}$ is the stop gradient operation for continuous training, $d_x, d_z$ are respectively the distances on the data and latent space and $\beta$ is set between $0.1$ and $2.0$ in the original proposal (Van Den Oord et al., 2017).

In our work, we explore a different model to learning discrete representations. Following VQ-VAE, we also consider $Z$ as a $M-$component latent embedding. On a $k^{th}$ sub-space (for $k \in \{1, \cdots, K\}$), we impose a Gaussian distribution parameterized by $\mu_k, \Sigma_k$ where $\Sigma_k$ is diagonal. We also endow $M$ discrete distributions over $\mathbf{C}^1, \ldots, \mathbf{C}^M$, sharing a common support set as the set of sub-spaces induced by $\{(\mu_k, \Sigma_k)\}_{k=1}^K$:

$$\mathbb{P}_{k,\pi^m} = \sum_{k=1}^{K} \pi_k^m \delta_{\mu_k}, \text{ for } m = 1, \ldots, M.$$

with the Dirac delta function $\delta$ and the weights $\pi^m \in \Delta_{K-1} = \{\alpha \geq \mathbf{0} : \|\alpha\|_1 = 1\}$ in the $(K-1)$-simplex. The probability a data point $z^m$ belongs to a discrete $k^{th}$ sub-space follows a $K-$way categorical distribution $\pi^m = [\pi_1^m, \cdots, \pi_K^m]$. In such a practical setting, the generative process is detailed as follows

1. For $m \in \{1, \cdots, M\}$,
   - Sample $k \sim \text{Cat}(\pi^m)$,
   - Sample $z^m \sim \mathcal{N}(\mu_k, \Sigma_k)$,
   - Quantize $\mu_k^m = Q(z^m)$,
2. $x = \psi_\theta([z^m]_{m=1}^M, [\mu_k^m]_{m=1}^M)$.

where $\psi$ is a highly non-convex function with unknown parameters $\theta$. $Q$ refers to the quantization of $[z^m]_{m=1}^M$ to $[\mu_k^m]_{m=1}^M$ defined as $\mu_k^m = Q(z^m)$ where $k = \operatorname{argmin}_k\, d_z\big(z^m; \mu_k\big)$ and $d_z = \sqrt{(z^m - \mu_k)^T \Sigma_k^{-1}(z^m - \mu_k)}$ is the Mahalanobis distance.

The backward map is defined via an encoder function $f_e$ and quantization process $Q$ as

$$\phi(x) = \big[f_e(x), Q(f_e(x))\big], \quad z = [z^m]_{m=1}^M = f_e(x), \quad [\mu_k^m]_{m=1}^M = Q(z).$$

The learnable parameters are $\{\pi, \mu, \Sigma, \theta\}$ with $\pi = [[\pi_k^m]_{m=1}^M]_{k=1}^K, \mu = [\mu_k]_{k=1}^K, \Sigma = [\Sigma_k]_{k=1}^K$.
Applying OTP-DAG to the above generative model yields the following optimization objective:

$$\min_{\pi,\mu,\Sigma,\theta} \quad \mathbb{E}_{X\sim\mathcal{D}}\bigg[c\big[X, \psi_\theta(Z, \mu_k)\big]\bigg] + \frac{\eta}{M} \sum_{m=1}^M \big[D_{\text{WS}}\big(P_\phi(Z^m), P(\widetilde{Z}^m)\big) + D_{\text{WS}}\big(P_\phi(Z^m), \mathbb{P}_{k,\pi^m}\big)\big]$$

$$+ \eta_r \sum_{m=1}^M D_{\text{KL}}\big(\pi^m, \mathcal{U}_K\big),$$

where $P_\phi(Z^m) := f_e^m \# P(X)$ given by the backward $\phi$, $P(\widetilde{Z}^m) = \sum_{k=1}^K \pi_k^m \mathcal{N}(\widetilde{Z}^m | \mu_k, \Sigma_k)$ is the mixture of Gaussian distributions. The copy gradient trick (Van Den Oord et al., 2017) is applied throughout to facilitate backpropagation.

The first term is the conventional reconstruction loss where $c$ is chosen to be mean squared error. Minimizing the second term $D_{\text{WS}}\big(P_\phi(Z^m), P(\widetilde{Z}^m)\big)$ forces the latent representations to follow the Gaussian distribution $\mathcal{N}(\mu_k^m, \Sigma_k^m)$. Minimizing the third term $D_{\text{WS}}\big(P_\phi(Z^m), \mathbb{P}_{k,\pi^m}\big)$ encourages every $\mu_k$ to become the clustering centroid of the set of latent representations $Z^m$ associated with it. Additionally, the number of latent representations associated with the clustering centroids are proportional to $\pi_k^m, k = 1, ..., K$. Therefore, we use the fourth term $\sum_{m=1}^M D_{\text{KL}}\big(\pi^m, \mathcal{U}_K\big)$ to guarantee every centroid is utilized.

**Training Configuration.** We use the same experiment setting on all datasets. The models have an encoder with two convolutional layers of stride 2 and filter size of $4 \times 4$ with ReLU activation, followed by 2 residual blocks, which contained a $3 \times 3$, stride 1 convolutional layer with ReLU activation followed by a $1 \times 1$ convolution. The decoder was similar, with two of these residual blocks followed by two de-convolutional layers. The hyperparameters are: $D = M = 64, K = 512, \eta = 1e-3, \eta_r = 1.0$, batch size of 32 and 100 training epochs.

**Evaluation Metrics.** The evaluation metrics used include (1) **SSIM:** the patch-level structure similarity index, which evaluates the similarity between patches of the two images; (2) **PSNR:** the pixel-level peak signal-to-noise ratio, which measures the similarity between the original and generated image at the pixel level; (3) feature-level **LPIPS** (Zhang et al., 2018), which calculates the distance between the feature representations of the two images; (4) the dataset-level Fr'echlet Inception Distance **(FID)** (Heusel et al., 2017), which measures the difference between the distributions of real and generated images in a high-dimensional feature space; and (5) **Perplexity:** the degree to which the latent representations $Z$ spread uniformly over $K$ sub-spaces i.e., all $K$ regions are occupied.

**Qualitative Examples.** We first present the generated samples from the CelebA dataset using Image transformer (Parmar et al., 2018) as the generative model. From Figure 9, it can be seen that the discrete representation from the our method can be effectively utilized for image generation with acceptable quality.

We additionally show the reconstructed samples from CIFAR10 dataset for qualitative evaluation. Figure 10 illustrate that the reconstructions from OTP-DAG have higher visual quality than VQ-VAE. The high-level semantic features of the input image and colors are better preserved with OTP-DAG than VQ-VAE from which some reconstructed images are much more blurry.

**Additional Experiment.** We additionally investigate a recent model called SQ-VAE (Takida et al., 2022) proposed to tackle the issue of codebook utilization. Table 8 reports the performance of SQ-VAE in comparison with our OTP-DAG. We significantly outperform SQ-VAE on Perplexity, showing that our model mitigates codebook collapse issue more effectively, while compete on par with this SOTA model across the

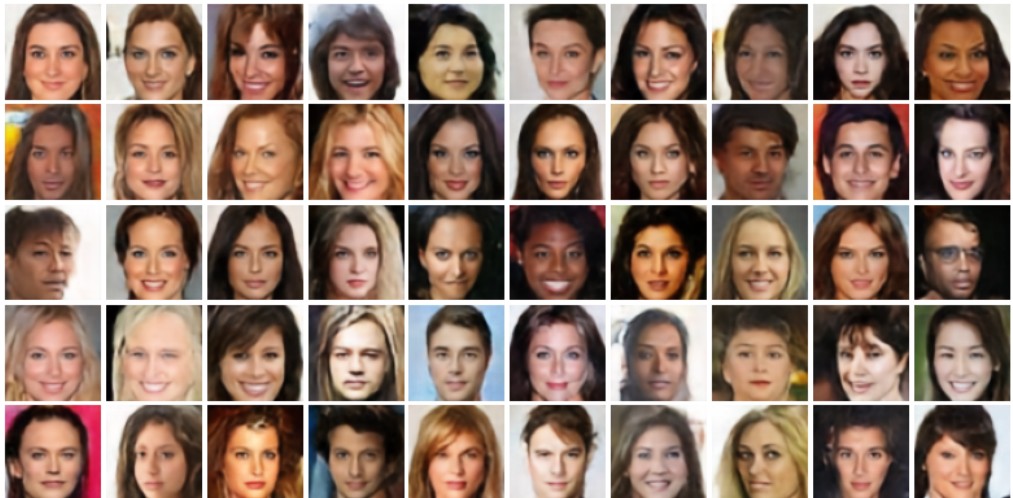

Figure 9: Generated images from the discrete representations of OTP-DAG on CelebA dataset.

other metrics. It is worth noting that our goal here is not to propose any SOTA model to discrete representation learning, but rather to demonstrate the applicability of OTP-DAG on various tasks, particular problems where traditional methods such as EM, MAP or mean-field VI cannot simply tackle.

Table 8: Quality of image reconstructions

| Dataset | Method | Latent Size | SSIM ↑ | PSNR ↑ | LPIPS ↓ | rFID ↓ | Perplexity ↑ |
|---------|--------|-------------|--------|--------|---------|--------|--------------|
| CIFAR10 | **SQ-VAE** | $8 \times 8$ | 0.80 | **26.11** | 0.23 | **55.4** | 434.8 |
| | **OTP-DAG** (Ours) | $8 \times 8$ | 0.80 | 25.40 | 0.23 | 56.5 | **498.6** |
| MNIST | **SQ-VAE** | $8 \times 8$ | **0.99** | **36.25** | 0.01 | **3.2** | 301.8 |
| | **OTP-DAG** (Ours) | $8 \times 8$ | 0.98 | 33.62 | 0.01 | 3.3 | **474.6** |
| SVHN | **SQ-VAE** | $8 \times 8$ | **0.96** | **35.35** | **0.06** | **24.8** | 389.8 |
| | **OTP-DAG** (Ours) | $8 \times 8$ | 0.94 | 32.56 | 0.08 | 25.2 | **462.8** |
| CELEBA | **SQ-VAE** | $16 \times 16$ | 0.88 | **31.05** | 0.12 | 14.8 | 427.8 |
| | **OTP-DAG** (Ours) | $16 \times 16$ | 0.88 | 29.77 | **0.11** | **13.1** | **487.5** |

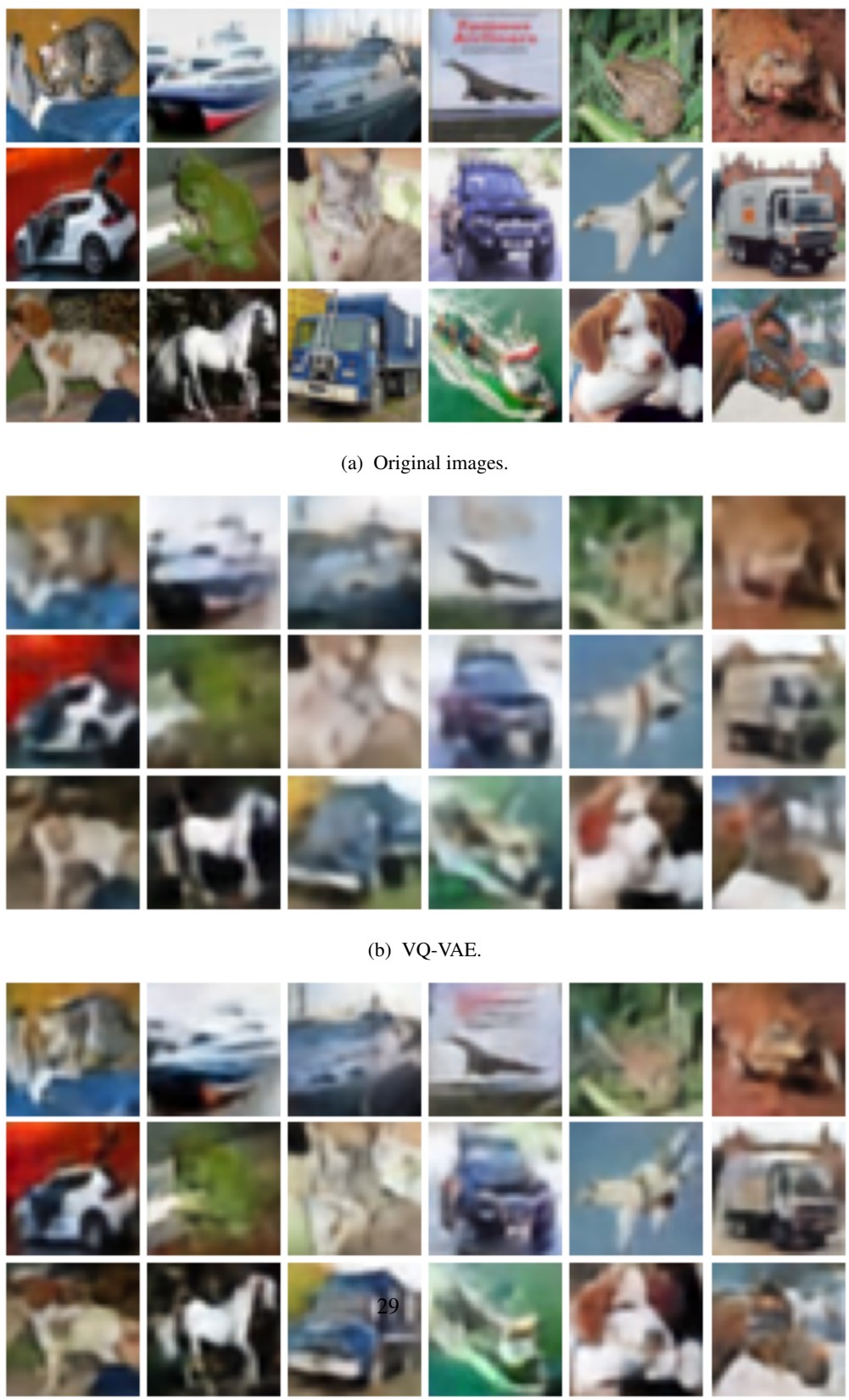

(a) Original images.

(b) VQ-VAE.

(c) OTP-DAG.

Figure 10: Random reconstructed images from CIFAR10 dataset.
.