# OpenReview forum: "Learning Directed Graphical Models with Optimal Transport"
_ICLR.cc/2024/Conference — Submitted to ICLR 2024_

### Official Review · Reviewer_zaHP · 2023-10-31

**Soundness:** 3 good
**Presentation:** 2 fair
**Contribution:** 2 fair
**Rating:** 3
**Confidence:** 4

**Summary:**

The paper presents a general approach to estimating the parameters of directed graphical models using Optimal Transport. The key idea is to relate minimization of reconstruction error with minimization of cost in OT. The approach has similarities to autoencoders and WAEs specifically. Three experimental evaluations are conducted showing that the approach is comparable to existing methods for parameter estimation in topic models, HMMs, and discrete representation learning.

**Strengths:**

- Optimal transport is a popular approach and there is broad interest in applications to various problems.
- Parameter estimation in DAGs is a classic topic for which there is interest.

**Weaknesses:**

I find it hard to understand what the contribution of the paper is:
-- There are a number of relatively empty claims (a partial list is included below), which are a distraction.
-- The introduction seems relatively unrelated to the main contribution of the text. For example, the second paragraph contains related approaches that aren't systematically introduced in the main text. Why are we talking about these?
-- The contributions include: "showing that minimizing the transport cost is equivalent to minimizing the reconstruction error between the observed data and the model generation". This is not a clear statement of what is new. The authors themselves note that this has been used in VAEs for example, and that is not the only place.
-- The OT approach is introduced mathematically, but I didn't find useful insight into how it was or was not related to other approaches (aside from being OT).
-- The experimental setup is introduced before baselines. Any informative experimental setup should be chosen to expose interesting contrasts with baselines. The logic doesn't make sense.

Detailed comments:
- The first sentence of the abstract isn't great. It is more informative to say what the problem is than to say it is a long standing challenge.
- "While existing learning methods are fundamentally based on likelihood maximization, here we offer a new view of the parameter learning problem through the lens of optimal transport." What is the new view? Is the intent to contrast OT with maximum likelihood?
- "Here we characterize them between two extremes." What is them?
- "As the complexity of the graph increases, despite the current advancements, parameter estimation in VI becomes less straightforward and computationally challenging." More details would be helpful here.
- "We present an entirely different view " In what way?
- I don't really understand the point of Figure 1.
- "laying a foundation stone for a new paradigm of learning and, potentially, inference of graphical models." What does this mean?
- "alternative line of thinking about parameter learning" What does this mean? Also, "Diverging from the existing frameworks"
- "We present theoretical developments showing that minimizing the transport cost is equivalent to minimizing the reconstruction error between the observed data and the model generation." Isn't this result already in the literature in multiple places? (It is fairly straightforward to show.)
- "While the formulation in Eq. (1) is not trainable," What does this mean?
- "for solving it efficiently " what is it?
- "Instead of achieving state- of-the-art performance on specific applications," Please say more. Why not?
- "We conclude with a more challenging setting: (3) Dis- crete Representation Learning (Discrete RepL) that cannot simply be solved by EM or MAP (maximum a posteriori). It in fact invokes deep generative modeling via a pioneering development called Vector Quantization Variational Auto-Encoder (VQ-VAE, Van Den Oord et al., 2017). " Please explain: what is challenging, why should we care about this model?
- "except the special setting of Discrete RepL" What makes this special?
- Not sure how I feel about the baselines appearing after the experimental setup. Shouldn't the setup be used to assess against the baselines?
- For LDA why isn't Gibbs sampling a baseline?
- The future research section is not particularly informative.

Overall the result is that I don't find the contribution clear or compelling. I believe there is something interesting here; however, I think there is a fair amount of work in repackaging (including possible new results) to have a compelling contribution.

**Questions:**

Please see the above comments. Perhaps the most important question to answer would be: What is the main contribution of the paper?

---

> ### Author Response · Authors · 2023-11-20
> **Contribution of Our Work**
>
> Much as we deeply appreciate the reviewer's time, we are perplexed by the reviewer's difficulty in understanding the contribution of our paper, which we believe has been clearly elucidated in the very first lines of the abstract and introduction. Let us take this opportunity to clarify the research problem and contribution.
>
> **1. Contribution**
>
> We here address the fundamental issue of learning parameters in directed graphical models from incomplete data. Referring to the historical development of graphical learning (summarized in Figure 1), methods can be characterized towards two extremes:
>
> (1) At one extreme, one must make restrictive assumptions about the tractability of the posterior distribution (e.g., EM), the structure (e.g., mean-field approximations) or the model class (e.g., using conjugate exponential families).
>
> (2) At the other extreme, there are black-box inference methods, notably VI. Recall that VI is an inference method where learning is treated as a by-product and model parameters (denoted by $\theta$) are cast as global latent variables, along with the local latent variables (denoted by $z$). Given observed data $\mathcal{D}$, the posterior $p(z, \theta \vert \mathcal{D})$ is generally intractable, thus less straightforward in a complex graph without making mean-field assumptions. A common strategy is to marginalize over local latent variables, often done analytically and not always feasible. Another strategy is to collapse all the parameters into a single latent variable to apply amortized inference, which makes little use of the graphical structure.
>
> It is well known that existing methods, including EM and VI are fundamentally based on likelihood maximization, which is equivalent to minimizing the KL divergence between the model and empirical data distribution. Our method indeed contrast OT with maximum likelihood by minimizing the Wasserstein distance between the model and data distribution, as opposed to KL divergence. This is what we meant by claiming that this is an alternative line of thinking that views parameter learning as an OT problem where the Wasserstein formulation follows suite. Given the above limitations of existing methods, we would like to contribute a versatile learning algorithm that can be straightforwardly applied to any graphical structure and variable types.
>
> **2. Methodology**
>
> To the comment *"Isn't this result already in the literature in multiple places? (It is fairly straightforward to show.)"*:
>
> A related problem formulation can be found in the WAE paper that presents the theoretical result for a simple graph of 1 observed node and 1 hidden node. We generalize this result to learning an arbitrary graph of multiple nodes and our formulation thus inherits some good properties, as discussed in Appendix D. While the theoretical result might be straightforward to some, our contribution also involves transforming it into a practical and scalable algorithm that has been empirically shown to work effectively on a wide range of problems. To the best of our knowledge, the generalization result for graphical learning has not been presented elsewhere.
>
> **3. Experimentation**
>
> To the comment *"Any informative experimental setup should be chosen to expose interesting contrasts with baselines. The logic doesn't make sense."*.
>
> We find this comment quite confusing, since our experimental setup indeed aims to expose the contrasts with the baselines. Concretely, we evaluate the effectiveness of our method on both parameter recovery and downstream tasks.
>
> Our evaluation has been conducted on a wide range of models with different types of latent variables (continuous and discrete) and for different types of data (texts, images, and time series). We begin with a simple and popular task of topic modeling where traditional methods like EM or SVI can solve. We then consider learning HMMs, which remains fairly challenging, with known optimization/inference algorithms (e.g., Baum-Welch algorithm) often too computationally costly to be used in practice. For learning discrete representations, despite the simple graph, the true generative function is unknown and it is often approximated with a deep neural network with a number of parameters that can scale up to millions. Solving those problems with EM or MAP would be both expensive and generally intractable.
>
> We believe that the models used in our experiments are representative and demonstrate our frameworks' applicability well. With the versatility of our framework, it has great potential to be applied to more complex models.

---

### Official Review · Reviewer_7CB5 · 2023-10-31

**Soundness:** 2 fair
**Presentation:** 3 good
**Contribution:** 3 good
**Rating:** 6
**Confidence:** 3

**Summary:**

This paper proposes the optimal transport framework for learning parameters of probabilistic models, called OTP-DAG. Authors show that minimizing the transport cost is equivalent to minimizing the data reconstruction error, with detailed proofs. Experiments on three models validate the effectiveness of the proposed OTP-DAG method in terms of both data reconstruction and parameter estimation. This validates the scalable and versatile feature of the OTP-DAG and implies the potential of its practicality.

**Strengths:**

* The proposed OPT-DAG is derived step-by-step with solid math proofs, makes the method clear and intuitive.
* The idea is versatile and scalable to different models and applications, which means broad practicality of the model
* The idea of combining the optimal transport and parameter learning of probabilistic models is interesting to me.

**Weaknesses:**

* Some experiments are not that supportive and need to be improved. See questions.
* Latent variables inference is less discussed and compared in this paper.

**Questions:**

* Page 2 line 8, "where the data distribution is the source and the true model distribution is the target". Do you mean that the data distribution is $p_{\theta}(x)$ and the true model distribution is $p_{\theta_{\text{true}}}(x)$? But VI is minimizing the KL divergence between the two posterior distributions, which is not the data distribution. These sentences are a bit confusing.
* For 4.1, could you please provide a table showing e.g. the mean error of the estimated parameters w.r.t. the true parameters from different methods? I know there are similar reports in Table 4 in the appendix, but could you find a problem where the estimated parameter is the best among all baselines?
* For 4.2, the synthetic dataset simulated from HMM is not credible to me. Why not really sample hidden $Z_t$ from the Markov process, but specify the state-changing points? Also, have you tried other settings (other true parameter sets, randomly sampled from a hyperprior distribution), and report metrics with means and error bars? In this way, we can be convinced that the proposed method is significantly better than others. Besides, why not also learn $p$, the transition probabilities? Since the traditional EM algorithm can also learn the transition matrix (as a learnable parameter) of HMM. If the proposed model is not even comparable to EM, this example application is only acceptable but not supportive.

In summary, the score of 5 is not from the method part but from the experiments I mentioned above. I would like to increase the score, if authors are able to provide some extra competitive results from OTP-DAG with enough randomness of the choice of the true parameter when generating synthetic datasets.

---

> ### Author Response · Authors · 2023-11-20
> **Additional Experiments**
>
> We truly appreciate the reviewer's time and interest in our work. We address the reviewer's questions in the following.
>
> **Page 2 line 8, "where the data distribution is the source and the true model distribution is the target".**
>
> It is precisely between the data distribution $P_d(X)$ and the model distribution $P_{\theta}(X)$.
>
> **But VI is minimizing the KL divergence between the two posterior distributions, which is not the data distribution. These sentences are a bit confusing.**
>
> VI is indeed minimizing the KL divergence between the two posterior distributions. Referring to Eq. (10) (Appendix D, page 19), if $\mathcal{Q}$ contains all conditional probability distributions $\phi(Z \vert X)$, the VAE objective concurs with the negative marginal log-likelihood $-\mathbb{E}\_{P_{d}(X)} [\log P_{\theta}(X)]$.  To make the KL term tractable, the vanilla VAE uses a standard normal $P(Z)$ and restricts the variational distribution to a class of Gaussian distributions. Consequently, VAE is minimising an upper bound on the negative log-likelihood or, equivalently, on the KL divergence between the data and model distribution. Our claims are concerned with this connection.
>
> **Application 4.1**
>
> The following table reports the mean absolute error ($\times 10^2$) of the estimated parameters w.r.t. the true parameters (best/second-best methods are highlighted in bold/italic). This additional result further demonstrates the consistent performance of our method. However, it can be seen that the evaluation in terms of distribution divergence/distance as reported in the main paper provides a more conclusive and numerically significant comparison of methods.
>
> | $K$ | $M$    | $N$ | OTP-DAG (Ours)    | EM                | SVI               | Prod LDA          |
> |-----|--------|-----|-------------------|-------------------|-------------------|-------------------|
> | 10  | 1,000  | 100 | *6.048 $\pm$ 0.068* | 6.355 $\pm$ 0.870 | **5.979 $\pm$ 0.371** | 6.375 $\pm$ 0.039 |
> | 20  | 5,000  | 200 | *1.823 $\pm$ 0.036* | 1.823 $\pm$ 0.068 | 1.831 $\pm$ 0.075 | **1.798 $\pm$ 0.007** |
> | 30  | 10,000 | 300 | **0.815 $\pm$ 0.010** | 0.824 $\pm$ 0.027 | *0.821 $\pm$ 0.008* | 0.829 $\pm$ 0.003 |
>
> **Application 4.2**
>
> Yes, our model can learn the transitional probability as well. Following the reviewer's comment, we conduct an additional experiment in which we create synthetic data with the hidden states now generated from the HMM described in Section 4.2. Here we treat transition probability $p$ as a learnable parameter in addition to Poisson rates $\lambda_{1:4}$.
>
> We further impose on the parameters the uniform distributions where
> * $\lambda_1 \sim  U(10,20)$
> * $\lambda_2 \sim  U(30,40)$
> * $\lambda_3 \sim U(50,60)$
> * $\lambda_4 \sim U(80,90)$
> * $p \sim U(0.80, 0.90)$
>
> We randomly generate $200$ datasets of $50,000$ observations each. For each dataset, we train the models for $50$ iterations with learning rate of $0.05$ at $5$ different initializations, while keeping the same experimental configuration as reported in Appendix E.2.
>
> We report mean error of the estimates of the parameters in the following.
>
> * Poisson rates:
> | $\lambda_1$: $0.380 \pm 1.107$
> | $\lambda_2$: $0.784 \pm 0.770$
> | $\lambda_3$ $1.653 \pm 1.243$
> | $\lambda_4$: $0.930 \pm 1.084$
>
> * Transition probability $p$: $0.019 \pm 0.011$.
>
> We additionally visualize the distribution of the estimates to show the alignment with the generative uniform distributions. The figure can be anonymously viewed here: https://anonymous.4open.science/r/OTP-7944/hmm_estimates.png
>
> The code has also been added to our anonymous public repository.
>
> We hope to have addressed your concerns. If there are any more questions, we would love to receive your feedback.

---

> > ### Comment · Reviewer_7CB5 · 2023-11-20
> > **Raised my score from 5 to 6**
> >
> > Thanks! These results are more convincing. I have raised my score to 6. Currently, I have no further questions, and I would like to wait for other reviewers to digest their responses.

---

### Official Review · Reviewer_Y8kc · 2023-11-03

**Soundness:** 3 good
**Presentation:** 3 good
**Contribution:** 2 fair
**Rating:** 5
**Confidence:** 3

**Summary:**

The authors propose to estimate the parameters of a Bayesian Network through minimization of the Wasserstein distance between the empirical distribution over observed variables and the marginal distribution of the observed variables of the model . They propose a method for computing this Wasserstein distance by introducing a collection of "reversed" kernels from observation to hidden variables.

**Strengths:**

The paper is clear.

**Weaknesses:**

It seems to me that gradient descent in equation 2 implies summing over all the parent nodes $PA_{X_O}$ , which seems very costly. If it is so, it is a limitation of the method. It would have been very nice to see how such a method compares to message passing algorithms for Bayesian Networks.

**Questions:**

How does the proposed gradient descent compare in terms of complexity with belief propagation?

---

> ### Author Response · Authors · 2023-11-19
> **Complexity of our method**
>
> We thank the reviewer for the feedback. Below are our responses to the reviewer's question.
>
> **Summing over all the parent nodes seems very costly.**
>
> Our algorithm is **not** costly, because (1) we do not perform inference during training, rather directly learn the parameters; (2) we support mini-batch training by leveraging amortized optimisation. Empirically our training time is less than half of that of batch methods like EM. In our experiment with hidden Markov models specifically, the model converges quickly after 50 iterations across most synthetic datasets, given a relatively long sequence of $200$ steps. This is more significantly efficient than the forwards-backwards (or Baum-Welch) algorithm (a form of message passing).
>
> **How does the proposed gradient descent compare in terms of complexity with belief propagation?**
>
> While our algorithm OTP-DAG does share the same ``forwards-backwards'' nature with belief propagation (BP), it is worth noting that our method focuses on parameter estimation (i.e., learning) rather than inference.
>
> In terms of complexity, the main difference is that OTP-DAG optimises the backward and forward distributions simultaneously via gradient descent, whereas standard BP computes messages recursively. The running time of message passing algorithms is generally exponential in the tree-width of the graph. Meanwhile, OTP-DAG is much less costly by characterising the local densities via the backward maps from the observed nodes to their parents, through which computation is localised and can be done in parallel.

---

> > ### Comment · Reviewer_Y8kc · 2023-11-20
> > **Reply**
> >
> > I would like to thank very much the authors for their reply. If I am not mistaken, BP serves, in a similar fashion to what authors propose, as one step for learning the parameters: it is the E step of the EM algorithm for graphical models.

---

> ### Author Response · Authors · 2023-11-21
> **How OTP-DAG is different**
>
> We thank the reviewer for the feedback.
>
> **1. To the reviewer's comment:** *"BP serves, in a similar fashion to what authors propose, as one step for learning the parameters: it is the E step of the EM algorithm for graphical models."*
>
> Our algorithm is fundamentally different from BP.
>
> First and most importantly, the focus of our work is on parameter estimation of directed graphical models, **NOT** inference.
>
> For the parameter learning task, it is true that one could use EM and perform BP in the E step (perhaps at the expense of computational cost due to the recursive nature of the algorithm). With this being said, it should essentially be comparing OTP-DAG with EM, not with BP.
>
> **2. The methodological differences with EM and other existing methods have been described in the paper. Let us provide another high-level perspective to once again clarify our novelty.**
>
> It is seen that when estimating the parameters, existing algorithms involve doing inference in some manner, e.g., explicitly via VI (as in the name) or iteratively in the E step of EM (either done analytically or with BP).
>
> This is what sets us apart. OTP-DAG does not perform inference. Through the OT formulation, our method directly learns the parameters. We do not explicitly compute the posterior marginal or conditional probabilities. Each backward distribution is constructed in the way that (1) satisfies the push-forward constraints and (2) we can sample from it to evaluate the reconstruction. Our algorithm thus avoids the computational complexity introduced by inference, which allows us to estimate the parameters more efficiently.
>
> Finally, it is natural to ask how OTP-DAG can do inference. This is indeed an interesting question that requires future exploration. As big fans of graphical models, we hope that our approach could potentially set the stage for solving large-scale inference and discovery problems.

---

### Meta-Review · Area_Chair_otpC · 2023-12-06

**Metareview:**

This paper addresses the issue of learning directed graphical models. They develop a method to do so using optimal transport ideas. The reviews for this paper were a bit mixed, but more fell below the threshold than above. The reviewers appreciated the topic and the efforts to bring optimal transport into the problem. But, substantial weaknesses were noted. One reviewer identified a mismatch between the claims and the evidence to support the claims. The clarity of the paper is lacking and a reviewer identified several sentences that were highly ambiguous, had pronouns with unclear antecedents, or lacked citations to support the indicative. I tend to agree that Figure 1 may not serve to clarify the paper and the authors may consider revising it given the space limitations of a conference paper. The author responses to to reviewers were considered in this metareview and it is my hope that the authors consider the feedback seriously in their revisions of this work.

**Justification For Why Not Higher Score:**

The reviewers raised many valid concerns. The clarity is lacking and the claims are not well supported by the experiments.

**Justification For Why Not Lower Score:**

N/A

---

### Decision · Program_Chairs · 2024-01-16

Reject